# Mutation of *Pseudomonas aeruginosa lasI/rhlI* diminishes its cytotoxicity, oxidative stress, inflammation, and apoptosis on THP-1 macrophages

Yanying Ren,[1,2] Xiaojuan You,[2,3] Rui Zhu,[2,3] Dengzhou Li,[2,3] Chunxia Wang,[2,3] Zhiqiang He,[2,3] Yue Hu,[3] Yifan Li,[2] Xinwei Liu,[2,3] Yongwei Li[2,3,4,5,6]

**ABSTRACT** The management of *Pseudomonas aeruginosa* (*P. aeruginosa*) infections presents a substantial challenge to clinics and public health, emphasizing the urgent need for innovative strategies to address this issue. Quorum sensing (QS) is an intercellular communication mechanism that coordinates bacterial activities involved in various virulence mechanisms, such as acquiring host nutrients, facilitating biofilm formation, enhancing motility, secreting virulence factors, and evading host immune responses, all of which play a crucial role in the colonization and infection of *P. aeruginosa*. The LasI/R and RhlI/R sub-systems dominate in the QS system of *P. aeruginosa*. Macrophages play a pivotal role in the host's innate immune response to *P. aeruginosa* invasion, particularly through phagocytosis as the initial host defense mechanism. This study investigated the effects of *P. aeruginosa*'s QS system on THP-1 macrophages. Mutants of PAO1 with *lasI/rhlI* deletion, as well as their corresponding complemented strains, were obtained, and significant downregulation of QS-related genes was observed in the mutants. Furthermore, the *ΔlasI* and *ΔlasIΔrhlI* mutants exhibited significantly attenuated virulence in terms of biofilm formation, extracellular polymeric substances synthesis, bacterial adhesion, motility, and virulence factors production. When infected with *ΔlasI* and *ΔlasIΔrhlI* mutants, THP-1 macrophages exhibited enhanced scavenging ability against the mutants and demonstrated resistance to cytotoxicity, oxidative stress, inflammatory response, and apoptosis induced by the culture supernatants of these mutant strains. These findings offer novel insights into the mechanisms underlying how the *lasI/rhlI* mutation attenuates cytotoxicity, oxidative stress, inflammation, and apoptosis in macrophages induced by *P. aeruginosa*.

**IMPORTANCE** *P. aeruginosa* is classified as one of the ESKAPE pathogens and poses a global public health concern. The QS system of this versatile pathogen contributes to a broad spectrum of virulence, thereby constraining therapeutic options for serious infections. This study illustrated that the *lasI/rhlI* mutation of the QS system plays a prominent role in attenuating the virulence of *P. aeruginosa* by affecting bacterial adhesion, biofilm formation, extracellular polymeric substances synthesis, bacterial motility, and virulence factors' production. Notably, THP-1 macrophages infected with mutant strains exhibited increased phagocytic activity in eliminating intracellular bacteria and enhanced resistance to cytotoxicity, oxidative stress, inflammation, and apoptosis. These findings suggest that targeted intervention toward the QS system is anticipated to diminish the pathogenicity of *P. aeruginosa* to THP-1 macrophages.

**KEYWORDS** *Pseudomonas aeruginosa*, quorum sensing, oxidative stress, inflammation, apoptosis, macrophages

Address correspondence to Xinwei Liu, 43154727@qq.com, or Yongwei Li, lyw@hactcm.edu.cn.

The authors declare no conflict of interest.

See the funding table on p. 22.

*P*seudomonas aeruginosa is an important opportunistic pathogen renowned for its remarkable environmental adaptability, rendering it highly predisposed to causing nosocomial infections such as ventilator-associated pneumonia, urinary tract infections, and bloodstream infections (1). Additionally, it serves as a major contributor to chronic infections. The escalating resistance of *P. aeruginosa* due to the extensive use of antibiotics poses a substantial challenge to clinical management (2). Therefore, it is imperative to explore innovative approaches for treating *P. aeruginosa* infections.

The quorum sensing (QS) system is a density-dependent signal communication system that is widely distributed in *P. aeruginosa* bacterial populations (3). It comprises three interrelated QS sub-systems of significant importance: Las, Rhl, and Pqs (4). The Las system occupies the apex of the regulatory network and is the primary responder to stimulation, exerting positive regulation over both the Rhl and Pqs systems. The *lasI* and *rhlI* genes encode two distinct acyl-homoserine lactone signaling molecules (AHLs), namely N-3-oxo-dodecanoyl-homoserine lactone (C12-HSL) and N-butyryl-homoserine lactone (C4-HSL) (5). AHLs are capable of binding to and activating their respective transcriptional activators, LasR and RhlR, thus forming a complex of signaling molecules and transcription activator proteins that ultimately regulate the expression of various virulence genes (6, 7). The Las system primarily regulates the synthesis of LasA protease, LasB elastase, alkaline protease, and exotoxin A, whereas the Rhl system is involved in the regulation of rhamnolipid and pyocyanin (8). The QS system plays a crucial role in regulating the complex interactions involved in biofilm development, bacterial adhesion, bacterial motility, and the production of various extracellular products among bacterial populations (9). These factors are intricately intertwined with the development of bacterial virulence, drug resistance, host immune evasion, and community behavior, all of which significantly contribute to refractory *P. aeruginosa* infections that pose a challenge to effective treatment strategies (1, 10). Hence, the intervention of *P. aeruginosa*'s QS system is expected to effectively modulate its pathogenicity and drug resistance, offering a novel approach to the treatment of *P. aeruginosa* infections. Currently, national and international research on the QS system of *P. aeruginosa* is mainly focused on the development of QS inhibitors. Nevertheless, there remains an inadequate comprehensive understanding of the regulatory mechanism of the QS system.

Macrophages play a crucial role in the innate immune response by actively phagocytosing and eliminating invading pathogens during the early stages of infection, particularly when bacterial load is insufficient to recruit neutrophils (11, 12). In this study, mutant and complemented strains of *lasI/rhlI* in the QS system of *P. aeruginosa* were obtained to evaluate the differences in transcriptomes and virulence phenotypes, particularly concerning biofilm formation, motility, and expression of virulence factors. Furthermore, the bacterial response and host response of THP-1 macrophages to *P. aeruginosa* were investigated to evaluate the effects of *lasI/rhlI* mutation on macrophages. This has facilitated our comprehension of the molecular mechanisms underlying the QS system and the pathogenic properties of *P. aeruginosa*, thus enabling us to develop effective strategies for its management.

## MATERIALS AND METHODS

### Materials

The strains and plasmids used in this investigation are detailed in Table S1, whereas the essential primers for gene knockout and complementation can be found in Table S2. Primer synthesis, DNA sequencing, and RNA-seq were conducted by Sangon Bioengineering Co., LTD (Shanghai, China).

## Construction of Δ*lasI* of PAO1

### Construction of the targeting plasmid pCVD442-Δ*lasI*::Gm

PAO1 mutants were generated based on Hamelo's method (13). The upstream and downstream homology arms of *lasI* gene were amplified from the genomic DNA of *P. aeruginosa* PAO1 with primers *lasI*-5F/5R and *lasI*-3F/3R, employing high-fidelity DNA polymerases (Takara Biomedical Technology Co., LTD, Peking, China). The gentamicin (Gm) resistance gene fragment was amplified from the plasmid pUC57-Gm, using primers Gm-F/R. Fusion PCR linked the upstream and downstream homology arms to the Gm gene, resulting in a fused fragment named Δ*lasI*::Gm. The Δ*lasI*::Gm was then cloned into the pCVD442-*Sma*I site of the suicide plasmid pCVD442 using T4 DNA ligase (Marrone Bio Innovations, Inc., CA, USA) following digestion with *Sma*I (Marrone Bio Innovations, Inc., CA, USA) at 30℃ for 2 h. Single colonies were obtained on a Luria-Bertani (LB) agar plate containing 50 µg/mL ampicillin (Amp) and 25 µg/mL Gm at 37℃ after electroporation into *Escherichia coli* DH5α λpir. The targeting plasmid pCVD442-Δ*lasI*::Gm was verified through DNA sequencing.

### Acquisition and conjugation experiment of donor bacterium β2155/pCVD442-Δ*lasI*::Gm

The *E. coli* β2155 diaminopimelic acid (DAP) auxotrophic strain requires a growth medium containing DAP. Electroporation was employed to introduce pCVD442-Δ*lasI*::Gm into *E. coli* β2155 competent cells. Single colonies grown on LB plates with 100 µg/mL Amp and 0.5 mM DAP were designated as donor strain β2155/pCVD442-Δ*lasI*::Gm while PAO1 served as the recipient strain. The donor and recipient strains were mixed in equal proportions and cultured at 30℃ and 220 rpm for 16 h for the conjugation experiment. Subsequently, the culture was incubated at 30℃ until positive colonies appeared on LB plates supplemented with 100 µg/mL Amp and 33 µg/mL Gm.

### Screening and identification of Δ*lasI*

Several positive colonies randomly selected from the conjugation experiment were mixed in an LB liquid medium. Reverse screening was conducted on LB plates containing 10% sucrose (with 33 µg/mL Gm but no NaCl). Positive colonies were identified via PCR using the internal primer *lasI*-in-F/R, and those showing negative results were validated using the external primer *lasI*-out-F/R. The colony with anticipated length was identified as the target strain Δ*lasI*.

## Construction of Δ*lasI*Δ*rhlI* of PAO1

The construction of Δ*lasI*Δ*rhlI* was similar to that of Δ*lasI*. The homology arms of *rhlI* and the apramycin (Apr) resistance gene fragment were amplified using the primers *rhlI*-5F/5R, *rhlI*-3F/3R, and Apr-F/R, respectively. Subsequently, fusion fragment Δ*rhlI*::Apr was obtained. After enzyme digesting, DNA ligating, and electroporating, the pCVD442-Δ*rhlI*::Apr recombinant plasmid was obtained on LB plates containing 50 µg/mL Amp and 100 µg/mL Apr. Conjugation experiments were performed between donor strain β2155/pCVD442-Δ*rhlI*::Apr and recipient strain Δ*lasI*. Similarly, LB plates with 10% sucrose (100 µg/mL Apr without NaCl) were used for reverse screening. The individual colonies obtained were verified using *rhlI*-inF/R and *rhlI*-outF/R. The colonies that tested negative with the internal primer but positive with the external primer were defined as Δ*lasI*Δ*rhlI*.

## Construction of complemented strains of Δ*lasI* (i.e., Δ*lasI*-Comp.) and Δ*lasI*Δ*rhlI* (i.e., Δ*lasI*Δ*rhlI*-Comp.)

The pRK415 plasmid carrying the tetracycline (TC) resistance gene was used as the complementing vector. The *lasI* fragment of PAO1 was amplified using primers *lasI*-F/R1, digested with *Xba*I and *EcoR*I, and then inserted into the corresponding restriction sites of the plasmid pRK415. Following ligation and transformation into *E. coli*

TOP10 competent cells, positive clones were selected on LB plates supplemented with 10 µg/mL TC to obtain the *lasI* complementing plasmid pRK415-*lasI*. Subsequently, electrotransformation of the plasmid pRK415-*lasI* into *E. coli* β2155 was performed, and positive clones were screened on LB plates containing 10 µg/mL TC and 0.5 mM DAP to acquire donor strain β2155/pRK415-*lasI*. Plasmid conjugation between the donor strain and recipient strain Δ*lasI* resulted in positive clones on LB plates containing 10 µg/mL TC. Finally, PCR (pRK415-F/R) and sequencing analysis were performed to confirm the successful transfer of the complementing plasmid into Δ*lasI*, generating the *lasI* gene complemented strain namely Δ*lasI-Comp.*.

The acquisition of Δ*lasI*Δ*rhlI-Comp.* followed a similar procedure as that for Δ*lasI-Comp.*. Specifically, the *lasI* and *rhlI* fragments were amplified from PAO1 using primers *lasI*-F/R2 and *rhlI*-F/R, respectively. These two fragments were then fused by PCR and digested with *Xba*I and *EcoR*I before being cloned into plasmid pRK415 to generate the complementing plasmid pRK415-*lasIrhlI*. After transformation, the donor strain β2155/pRK415-*lasIrhlI* was combined with the recipient strain Δ*lasI*Δ*rhlI* to achieve plasmid transfer. Finally, through positive clone screening and identification, the *lasI* and *rhlI* double gene complemented strain Δ*lasI*Δ*rhlI-Comp.* was obtained.

## Transcriptome analysis

### cDNA library construction and RNA-Seq

Total RNA was extracted from three replicates of PAO1, Δ*lasI*, and Δ*lasI*Δ*rhlI* to improve accuracy. After qualified detection, rRNA was removed from the total RNA, followed by fragmentation, cDNA synthesis, cDNA fragmentation modification, magnetic bead purification, fragmentation sorting, and library amplification. Ultimately, a specific RNA-Seq library was prepared using the Illumina HiSeq 2500 sequencing platform (Illumina Inc., CA, USA).

### Quality assessment of RNA-Seq data and analysis of reference genome alignment

FastQC 0.11.2 was used to evaluate the quality of raw sequencing data, and the clean reads were obtained after filtering out splice sequences and unknown base sequences. The clean reads were then aligned to the reference genome of *P. aeruginosa* PAO1 (NC_002516.2) using Bowtie2 2.3.2 (14). The gene expression levels in samples were calculated using the standardized Transcripts Per Million (TPM).

### Differential expression genes and enrichment analysis

Differential expression genes (DEGs) were identified using DEGseq 1.26.0, employing the criteria of |log2 (fold change)|> 1 and q-value <0.05. The log2 (fold change) represents a base 2 logarithm of the fold change, whereas the q-value indicates the adjusted *P*-value using false discovery rate (FDR) correction. For pathway enrichment, the Kyoto Encyclopedia of Genes and Genomes (KEGG) database was utilized to annotate pathways associated with the identified DEGs. Gene Ontology (GO) analysis classified and annotated functions of DEGs, identifying significantly enriched GO terms to determine their primary biological functions.

## Growth curve analysis

The strains of PAO1, Δ*lasI*, Δ*lasI*Δ*rhlI*, Δ*lasI-Comp.*, and Δ*lasI*Δ*rhlI-Comp.* were cultured in LB medium until they reached the logarithmic growth phase with an OD600 of 0.5. The cultures were then diluted to approximately $5 \times 10^5$ colony forming units (CFU)/mL and added to a 96-well microplate in equal volumes. The microplate was incubated at 37°C using a Feyond-A300 enzyme-linked immunosorbent analyzer (Allsheng Instrument Co., Ltd. Hangzhou, China), and OD600 was measured at hourly intervals over 24 h to generate a growth curve. The experiment was conducted independently in triplicate.

## RT-qPCR analysis

Total RNA was extracted using the TRIzol kit (Sangon Bioengineering Co., LTD., Shanghai, China), and the concentration and purity were determined by the measurement of OD260/280. Afterward, the total RNA was reverse transcribed into cDNA using the First Strand cDNA Synthesis Kit (Servicebio Co., LTD., Wuhan, China), followed by RT-qPCR analysis employing the SYBR Green qPCR Master Mix kit (Servicebio Co., LTD., Wuhan, China). The *16S rRNA* was chosen for *P. aeruginosa,* and β-actin served for THP-1 macrophages as an internal reference gene. The relative mRNA expression levels of target genes were calculated using the $2^{-\triangle\triangle Ct}$ method. The specific primers required for RT-qPCR are listed in Tables S3 and S4. The experiment was conducted independently in triplicate.

## Biofilm formation ability determination

### Bacterial adhesion

The PAO1, *ΔlasI*, *ΔlasIΔrhlI*, *ΔlasI-Comp..* and *ΔlasIΔrhlI-Comp.* with OD600 of 0.5 were continuously cultured for 16 h at 37℃ in sterile glass tubes containing 3 mL LB medium (at a ratio of 1:200). One test tube was collected every 2 h and washed three times with phosphate-buffered saline (PBS) to remove the planktonic bacteria. Bacteria attached to the tube wall were completely detached by adding 3 mL of 0.9% NaCl for oscillation. After gradient dilution, the bacterial suspensions were plated on LB plates and incubated at 37℃ overnight to perform CFU count. The experiment was conducted independently in triplicate.

### Crystal violet staining

The PAO1, *ΔlasI*, *ΔlasIΔrhlI*, *ΔlasI-Comp.*, and *ΔlasIΔrhlI-Comp.* were added to 96-well plates at a concentration of $5 \times 10^5$ CFU/mL and incubated at 37℃ for 24 h, 48 h, and 72 h, replacing the culture medium every 24 h. After incubation, the planktonic bacteria were gently removed with PBS three times. The biofilms were fixed in 3.7% methanol for 15 min, stained with 0.1% crystal violet for 20 min, washed with PBS three times, air dried, and then destained with 33% acetic acid for 15 min. The biofilm content was determined through the measurement of OD570. The experiment was conducted independently in triplicate.

### XTT reduction assay

The biofilm models were prepared as described above. After removing the planktonic bacteria, 200 µL of LB medium and 20 µL of XTT working solution (1 g/L XTT sodium salt:3.06 g/L phenazine methyl sulfate (PMS) = 200:1) were added and incubated at 37℃ for 3 h in the dark. OD490 was measured to assess bacterial survival in the biofilm. The relative survival rate was calculated using the formula: $Y = (OD_{experimental\ group} - OD_{blank\ group}) / (OD_{control\ group} - OD_{blank\ group}) \times 100\%$. The experiment was conducted independently in triplicate.

### Fluorescence microscope imaging

Three milliliters of PAO1, *ΔlasI*, *ΔlasIΔrhlI*, *ΔlasI-Comp.*, and *ΔlasIΔrhlI-Comp.* bacterial suspensions were inoculated into 12-well culture plates. Sterile cell slides were added to the wells and incubated at 37℃ for 24 h, 48 h, and 72 h with LB medium replaced daily. After removing the planktonic bacteria, the slides were fixed with 3.7% formaldehyde for 15 min and stained with 0.02% acridine orange (AO) for 15 min while avoiding light. The unbound dye was removed, and the slides were sealed and observed under an Eclipse Ti2 fluorescence microscope (Nikon Precision Machinery Co., LTD., Shanghai, China). The experiment was conducted in triplicate, and the mean fluorescence intensity was quantified.

## Scanning electron microscope (SEM) imaging

Sterile tinfoil (1 cm × 1 cm) with excellent electrical conductivity was chosen to prepare the biofilm model carriers, whereas other preparation methods remained unchanged as before (15). After fixation with 2.5% glutaraldehyde, the biofilms were dehydrated using an ethanol gradient (30%, 50%, 70%, 80%, 90%, and 100%), vacuum-dried, gold-coated, and examined using SEM (JSM-7800F, Japan).

## Determination of extracellular polymeric substances

### Extracellular protein assay

The protein concentration of bacterial culture supernatants was determined using bicinchoninic acid (BCA) assay (Beyotime Bioengineering Co., LTD., Shanghai, China). OD570 was measured to establish a standard curve and determine protein concentration. SDS-PAGE electrophoresis was subjected to separate the secreted extracellular protein of each strain, and the protein bands were visualized using Coomassie brilliant blue R250 staining. The experiment was conducted independently in triplicate.

### Exopolysaccharide assay

According to the fact that Congo red can firmly bind to polysaccharides, the bacterial suspensions of PAO1, $\Delta lasI$, $\Delta lasI\Delta rhlI$, $\Delta lasI$-Comp., and $\Delta lasI\Delta rhlI$-Comp. with OD600 of 0.5 were spotted onto an agar plate containing Coomassie brilliant blue (20 µg/mL) and Congo red (40 µg/mL). After incubation at 37℃ for more than 48 h, the color of the colonies and the surrounding agar was observed (16, 17). The content of exopolysaccharide was also determined by a modified phenol-sulfuric acid method (18). Briefly, these strains were grown in 12-well plates to induce mucus production and then precipitated at 4℃ for 1 h by adding the quintuple volume of anhydrous ethanol. After centrifugation at 8,000 rpm for 5 min, the precipitation was washed with 80% ethanol. Following another round of centrifugation, the precipitation was completely dissolved in heated ddH$_2$O at 100℃ for 5 min. Then, 100 µL of 5% phenol and 500 µL of H$_2$SO$_4$ were successively added to a 200 µL polysaccharide solution. OD488 was measured after heating at 95℃ for 20 min. The experiment was conducted independently in triplicate.

### Alginate assay

Following the carbazole-sulfuric acid colorimetric method of Farisa Banu (18), 200 µL of bacterial suspensions were added to 500 µL of B(OH)3-H$_2$SO$_4$ mixture (4:1). After swirling, 20 µL of 0.2% carbazole solution (prepared with ethanol) was added and incubated for 30 min at 55℃. The content of alginate was measured at OD530. The experiment was conducted independently in triplicate.

## Bacterial motility

The agar plate assays were employed to assess bacterial motility on or within the semi-solid medium. Swimming plates with 0.3% (wt/vol) agar, Swarming plates with 0.5% (wt/vol) agar, and Twitching plates with 1.0% (wt/vol) agar were prepared following M H Rashid's method (19). The bacterial suspensions (5 µL) were added to the surface of the Swimming plate and incubated upright at 30℃ overnight. The Swarming plate, containing 5 g/L D-glucose, was incubated at 30℃ for over 48 h. The Twitching plate was pierced with a microtip, followed by adding bacterial suspensions (2 µL) to the interface between the plate and agar, incubating at 37℃ for 24 h. The plate bottom was stained with 0.1% crystal violet for 10 min after discarding the ager. The diameter of the movement is proportional to the bacterial motility. The experiment was conducted independently in triplicate.

The bacterial motility in the liquid medium was captured using an Eclipse Ti2 inverted microscope equipped with a 60 × Plan APO NA 0.9 phase contrast objective, which was controlled by NIS-Elements D (AR version 5.11.01). A total of 51 frames were recorded at

0.153 s/frame for 7.8 s. Imaris.v9.9 was utilized to automatically track the 2D trajectory of bacteria, whereas their motility was characterized using instantaneous velocity and the average displacement on the X-axis relative to the initial position. The experiment was conducted independently in triplicate.

## Determination of virulence factors

### LasA protease activity determination

LasA is an endopeptidase that can cleave the pentaglycine cross-link in the peptidoglycan of *Staphylococcus aureus* (*S. aureus*), resulting in rapid lysis of *S. aureus* (20, 21). LasA protease activity was measured by its ability to lyse heat-killed *S. aureus* (22). *S. aureus* ATCC 29213 resuspended in 0.02 M Tris-HCl (pH 8.0) to OD600 of 1.0 ~ 1.2 and heated at 100℃ for 20 min. PAO1, *ΔlasI*, *ΔlasIΔrhlI*, *ΔlasI-Comp.*, and *ΔlasIΔrhlI-Comp.* filtered supernatants (20 µL) were added to heat-killed *S. aureus* suspension (180 µL) and cultured at 37℃. The dynamic changes of OD595 were monitored every 10 min for 2.5 h to record the lysis of *S. aureus*. The experiment was conducted independently in triplicate.

### LasB elastase assay

The activity of LasB elastase was measured using the Elastin Congo Red Assay (Sigma-Aldrich, SHBL9368) (23). The bacterial filtered supernatants (500 µL) were separately mixed with an equal volume ofElastin Congo red (ECR) buffer (10 mg ECR, 100 mM Tris-HCl, pH 7.5) and shaken for 18 h at 37℃. The reaction was terminated by adding 0.1 mL 0.12 Methylene diamine tetraacetic acid (EDTA). Insoluble ECR was removed, and the OD495 was measured. The LasB elastase activity was also determined using the EnzChek Elastase Assay Kit (Invitrogen, Carlsbad, USA, 2407850) (24). The relative fluorescence unit (RFU) was monitored dynamically at 10-min intervals for 2.5 h at 37℃ using the Feyond-A300 Enzyme Immunoassay Analyzer (Ex/Em = 470 nm/525 nm). Porcine pancreatic elastase, supplied with the kit, served as the positive control, and the reaction buffer as the negative control. The experiment was conducted independently in triplicate.

### Determination of pyocyanin

The bacterial suspensions were cultured in a Pseudomonas medium for detection of pyocyanin(PDP) medium (containing 10 mL/L glycerol, 20 g/L peptone, 1.4 g/L $MgCl_2$, and 10 g/L $K_2SO_4$, pH 7.2 ~ 7.4) for 5 days continuously at 37℃ and 200 rpm. At 24 h intervals, 3 mL of chloroform was added to an equal amount of the collected centrifuged supernatants for pyocyanin extraction. The lower blue organic layer was transferred to 1 mL of 0.2 M HCl, shaken vigorously, and allowed to stand, and the upper red aqueous phase was collected for measuring the OD520. Pyocyanin concentration (µg/mL) = OD520 × 17.072. The experiment was conducted independently in triplicate.

## Effects of *lasI/rhlI* mutation in the QS system of *P. aeruginosa* on THP-1 macrophages

### Cell culture

Human THP-1 monocytes were cultured in RPMI 1640 medium containing 10% fetal bovine serum (FBS) and 1% penicillin-streptomycin at 37℃ and 5% CO2. The AO/PI (acridine orange/propidium iodide) kit (Nexcelom Bioscience, USA, 2211230201) and Cellometer K2 Cell Viability Analyzer (Nexcelom Bioscience, USA) were used to determine cell viability above 90%. Phorbol-12-myristate-13-acetate (PMA) doses of 200 ng/mL was employed to induce the differentiation of THP-1 monocytes into adherent macrophages for 48 h (25).

## Intracellular killing assay

$1 \times 10^7$ cells/well were evenly distributed in 6-well plates and differentiated into macrophages. PAO1, *ΔlasI*, *ΔlasIΔrhlI*, *ΔlasI-Comp.*, and *ΔlasIΔrhlI-Comp.* strains were cultured in RPMI 1640 medium and normalized to a final concentration of $1 \times 10^7$ CFU/mL. THP-1 macrophages were infected with normalized bacterial suspensions at an multiplicity of infection (MOI) of 1:1 and incubated at 37°C for 4 h. Then, the medium was removed and washed with PBS three times, followed by incubation in a complete medium containing 50 µg/mL Gentamicin for an additional 2 h to eliminate extracellular bacteria. After another wash with PBS, 0.01% (vol/vol) Triton X-100 was added to lyse cells and release intracellular bacteria. The cell lysate was serially diluted and subjected to a CFU count to determine the clearance capacity of THP-1 macrophages against different strains. The experiment was conducted independently in triplicate.

## CCK-8 cell viability assay

The PAO1, *ΔlasI*, *ΔlasIΔrhlI*, *ΔlasI-Comp.*, and *ΔlasIΔrhlI-Comp.* with a concentration of $5 \times 10^5$ CFU/mL were cultured in RPMI 1640 medium for 24 h. After centrifugation at 4°C and 12,000 rpm for 15 min, the bacterial culture supernatants were filtered through a 0.22 µm filter before use. Following the standard of $1 \times 10^4$ cells/well in 96-well plates, the THP-1 macrophages were treated with the collected culture supernatants for 6 h, 12 h, and 24 h. Then, 10% CCK-8 working solution was added for a further 4 h, and absorbance at 450 nm was measured. Cell viability = ($A_{experimental\ group}$ - $A_{blank\ group}$) / ($A_{control\ group}$ - $A_{blank\ group}$) × 100%. The experiment was conducted independently in triplicate.

## Reactive oxygen species detection

THP-1 macrophages were allowed to adhere to a sterile cell slide in a 12-well plate at a density of $1 \times 10^7$ cells/well. Subsequently, a 3.3 µM DCFH-DA fluorescent probe was loaded onto cells at 37°C for 20 min in the dark using the reactive oxygen species (ROS) assay kit (Beyotime Biotechnology Co., LTD., Shanghai, China), and the unloaded probe was removed with RPMI 1640 medium. After being treated with bacterial culture supernatants for a duration of 2 h, the intracellular ROS levels were observed using an inverted fluorescence microscope. Additionally, the cells loaded and treated *in situ* were collected, and the mean fluorescence intensity was measured to assess ROS content using a flow cytometer (BD FACSCalibur, USA). The experiment was conducted independently in triplicate.

## Protein carbonyl content detection

$1 \times 10^7$ cells/well of THP-1 macrophages were exposed to bacterial culture supernatants for 4 h to assess protein oxidative damage using the protein carbonyl content detection kit (Sangon Bioengineering Co., LTD, Shanghai, China). The protein carbonyl undergoes a reaction with 2,4-dinitrophenylhydrazine to yield red 2,4-dinitrophenylhydrazone, exhibiting a distinctive absorption peak at 370 nm. Protein carbonyl content (µmol/mg protein) = ($A_{experimental\ group}$-$A_{blank\ group}$) /17.6/sample protein concentration. The experiment was conducted independently in triplicate.

## Inflammatory cytokines analysis

$5 \times 10^5$ cells/well of THP-1 macrophages were incubated with bacterial culture supernatants for 24 h. The medium was collected to quantify the secretion of inflammatory cytokines, including IL-1α, IL-1β, IL-6, IL-10, IL-12, and TNF-α using ELISA kits (Sangon Bioengineering Co., LTD, Shanghai, China). The experiment was conducted independently in triplicate.

## Terminal deoxynucleotidyl transferase-dUTP nick-end labeling (TUNEL) analysis

The TUNEL kit (Elabscience Biotechnology Co., Ltd, Wuhan, China) was employed to detect DNA strand breaks. $5 \times 10^5$ cells/well of THP-1 macrophages were induced to

adhere to the cell slides and treated with bacterial culture supernatants for 4 h. The cells were fixed with 4% paraformaldehyde at 4℃ for 2 h, followed by permeabilization with 0.2% Triton X-100 at 37℃ for 10 min. Subsequently, the cells were stained with fluorescein isothiocyanate (FITC) at 37℃ for 1 h and 4′,6-diamidino-2-phenylindole (DAPI) at room temperature for 5 min in darkness. For positive controls, the DNA was cleaved using DNase I to expose the 3′-OH end. Finally, the slides were sealed and observed under an inverted fluorescence microscope. The experiment was conducted independently in triplicate.

### Apoptosis detection

Apoptosis was detected using Annexin V-FITC/PI kit (Elabscience Biotechnology Co.,Ltd, Wuhan, China). $1 \times 10^6$ cells/well of THP-1 macrophages were induced to differentiate into macrophages. After being exposed to bacterial culture supernatants for 4 h, the adherent cells were harvested and resuspended in 100 µL binding buffer containing Annexin V-FITC and PI reagents (2.5 µL each) at room temperature for 20 min in darkness. Upon addition of another 400 µL binding buffer, apoptosis was promptly assessed on a flow cytometer. The experiment was conducted independently in triplicate.

### Western blot

The THP-1 macrophages were treated with bacterial culture supernatants for 6 h, and the proteins were extracted using RIPA lysis buffer (containing 1 mM PMSF) (Solarbio Technology Co., LTD, Beijing, China). After loading 10 µg of standardized proteins onto SDS-PAGE and transferring to polyvinylidene fluoride (PVDF) membranes, the PVDF membrane was blocked with 5% skim milk for 2 h at room temperature and incubated with iNOS, COX-2, IκBα, Phospho-IκBα, NF-κB p65, Phospho-NF-κB p65, SOD2, GPX4, Bax, Bcl-2, and GAPDH (Proteintech Biotechnology Co., LTD, Wuhan, China) polyclonal antibodies overnight at 4℃. horseradish peroxidase (HRP)-conjugated Goat Anti-Rabbit IgG secondary antibody (Proteintech Biotechnology Co., LTD, Wuhan, China) was incubated at room temperature for 1 h. The ECL detection kit (Sangon Bioengineering Co., LTD, Shanghai, China) was utilized to image protein bands in Tanon 5200 luminescence imaging system (Tanon Life Science Co., LTD, Shanghai, China). The experiment was conducted independently in triplicate.

### Statistical analysis

IBM SPSS Statistics 21.0 and Graph Pad Prism 9.0 were used for statistical analysis and graphing. Differences between multiple groups were compared using ANOVA and Tukey's multiple-comparison test. The results are presented as mean ± standard deviation (M ± SD) of at least three independent experiments. $P < 0.05$ was considered the level of significance. *, $P < 0.05$; **, $P < 0.01$; ***, $P < 0.001$; ****, $P < 0.0001$; ns, no significance.

## RESULTS

### Effect of *lasI*/*rhlI* mutation on the transcriptome of *P. aeruginosa*

The mutants of Δ*lasI* and Δ*lasI*Δ*rhlI* were successfully constructed, and the details were presented in Fig. S1 and S2. The expression patterns of DEGs (fold change > 2, q-value <0.05) of Δ*lasI* and Δ*lasI*Δ*rhlI* compared with PAO1 are shown in Fig. S3. The volcano plots reveal that following the knockout of *lasI* in PAO1, 1,026 genes were significantly upregulated while 270 genes were downregulated. In contrast, the knockout of both *lasI* and *rhlI* in PAO1 resulted in 994 upregulated genes and 208 downregulated genes (Fig. S3A and B). Focusing on the significantly downregulated DEGs across different comparison groups, the Venn diagram showed that Δ*lasI* and Δ*lasI*Δ*rhlI* shared 175 common downregulated DEGs, which were enriched for key regulatory genes in the QS system and virulence genes regulated by the QS system (Fig. S3C). Table 1 lists the downregulated DEGs in Δ*lasI*Δ*rhlI* compared with PAO1.

**TABLE 1** Downregulated DEGs and pathways of ΔlasIΔrhlI compared with PAO1[a]

| Gene ID | Gene name | Gene description | Pathway | Log2 (FoldChange) | q-Value |
|---|---|---|---|---|---|
| PA1432 | lasI | Acyl-homoserine-lactone synthase | Quorum sensing | −19.9670517215 | 0 |
| PA1430 | lasR | Transcriptional regulator | Quorum sensing | −1.233340834227 | 2.61690524428E-13 |
| PA3476 | rhlI | Acyl-homoserine-lactone synthase | Quorum sensing | −3.40863486068 | 0 |
| PA3477 | rhlR | Transcriptional regulator | Quorum sensing | −1.60463170241 | 3.91215565567E-257 |
| PA0996 | pqsA | Anthranilate—CoA ligase | Quorum sensing | −3.78096815563 | 3.48791708832E-300 |
| PA0997 | pqsB | Hypothetical protein | Quorum sensing | −2.93108006118 | 2.61861228135E-156 |
| PA0998 | pqsC | Hypothetical protein | Quorum sensing | −2.74723392962 | 6.56163291159E-126 |
| PA0999 | pqsD | 3-Oxoacyl-ACP synthase | Quorum sensing | −1.00673982818 | 2.00276795465E-49 |
| PA1000 | pqsE | Thioesterase PqsE | Quorum sensing | −1.086453468162 | 1.54477622917E-33 |
| PA1003 | pqsR | Transcriptional regulator MvfR | Quorum sensing | −1.047749050634 | 0.00233064335052 |
| PA2587 | pqsH | 2-Heptyl-3-hydroxy-4(1H)-quinolone synthase | Quorum sensing | −1.64213425397 | 1.39715119297E-168 |
| PA3479 | rhlA | Rhamnosyltransferase subunit A | Quorum sensing | −4.97060436644 | 0 |
| PA3478 | rhlB | Rhamnosyltransferase subunit B | Quorum sensing | −4.58068784562 | 0 |
| PA1130 | rhlC | Rhamnosyltransferase | Quorum sensing | −2.31958669047 | 2.29979491149E-250 |
| PA1871 | lasA | Protease LasA | Quorum sensing | −3.73361327223 | 0 |
| PA3724 | lasB | Elastase LasB | Quorum sensing | −1.39690736449 | 2.23965505673999E-159 |
| PA1249 | aprA | Alkaline metalloproteinase | Quorum sensing | −3.27891644501 | 0 |
| PA2570 | lecA | PA-I galactophilic lectin | Quorum sensing | −1.90853095399 | 0.00377469449885 |
| PA3361 | lecB | Fucose-binding lectin PA-IIL | Quorum sensing | −1.90147709942 | 7.79004665852E-40 |
| PA3064 | pelA | Hypothetical protein | Biofilm formation | −1.4091991826 | 0.0040919918262 |
| PA3063 | pelB | Pellicle/biofilm biosynthesis protein PelB | Biofilm formation | −1.114016260292 | 0.00122422028102 |
| PA4209 | phzM | Phenazine-specific methyltransferase | Phenazine biosynthesis | −1.32275469467 | 2.03581E-06 |
| PA4217 | phzS | Hypothetical protein | Phenazine biosynthesis | −1.10371833382 | 1.9992196114E-44 |

[a]Note: The value "0" represents the q-value, value that is quite low and infinitely approaching 0.

The top 12 significantly enriched biological metabolic pathways identified through KEGG enrichment of the downregulated genes exhibited similar results for both ΔlasI and ΔlasIΔrhlI, including quorum sensing, biofilm formation, phenazine biosynthesis, metabolic pathways, degradation of aromatic compounds, biosynthesis of antibiotics, microbial metabolism in different environments, cyan amino acid metabolism, and benzoate degradation pathways among others (Fig. S3D and E). The top 12 enriched GO terms for downregulated genes include secondary metabolite biosynthesis, cellular catabolic, protein transport by Sec complex, signal transduction, protein secretion by the type VI and type II secretion system, single-species biofilm formation, extracellular space, proteolysis, transferase activity, bacterial-type flagellum-dependent swarming motility, and phenazine biosynthesis (Fig. S3F and G).

## Validation of QS-related gene expression by RT-qPCR

The 24 h growth curve indicated that the knockout and complementation of lasI and rhlI of the QS system had no significant impact on bacterial growth. PAO1, ΔlasI, ΔlasIΔrhlI, ΔlasI-Comp., and ΔlasIΔrhlI-Comp. all displayed typical growth patterns with an initial delay from 0 ~ 4 h followed by logarithmic proliferation from 4 ~ 14 h and stable growth thereafter (Fig. S4A). However, the mRNA expression levels of key regulatory genes and virulence genes showed a consistency between their expression patterns and those observed in the transcriptome analysis. Knocking out lasI significantly downregulated lasR, rhlI, rhlR, pqsA, and pqsR. Simultaneous loss of both lasI and rhlI further enhanced the downregulation compared with PAO1 and the complemented strains (Fig. S4B) ($P <$ 0.05). The significantly downregulated virulence genes included alginate synthesis genes algD and algR, exopolysaccharide pslA and pelA, type IV pilus pilA, type B flagella filC, protease lasA, elastase lasB, alkaline protease aprA, rhamnose lipotransferase subunit rhlA and rhlB, as well as pyocyanin-modifying gene phzM (Fig. S4C and D) ($P < 0.05$).

Among these, the most pronounced decrease was observed in *lasB*, followed by *pslA* and *lasA*.

## Mutation of *lasI*/*rhlI* reduces the bacterial adhesion and biofilm formation of *P. aeruginosa*

The QS system regulates the adhesion ability of *P. aeruginosa*, which is a prerequisite for biofilm formation. By counting viable bacteria adhering to the walls and bottoms of the test tubes continuously for 16 h, we observed a significant reduction in the number of viable bacteria adhered with *ΔlasI* and *ΔlasIΔrhlI*, and the complemented strains showed a marked increase in adherent bacteria (Fig. 1A) (*P* < 0.05). Furthermore, the biofilm quantification showed a gradual increase in the levels of PAO1 over 72 h. However, the biofilm biomass was significantly reduced in *ΔlasI* and reached its lowest level in *ΔlasIΔrhlI* (Fig. 1B) (*P* < 0.05). Additionally, the XTT reduction assay also showed a significant reduction in bacterial survival within the biofilm following *lasI* and *rhlI*

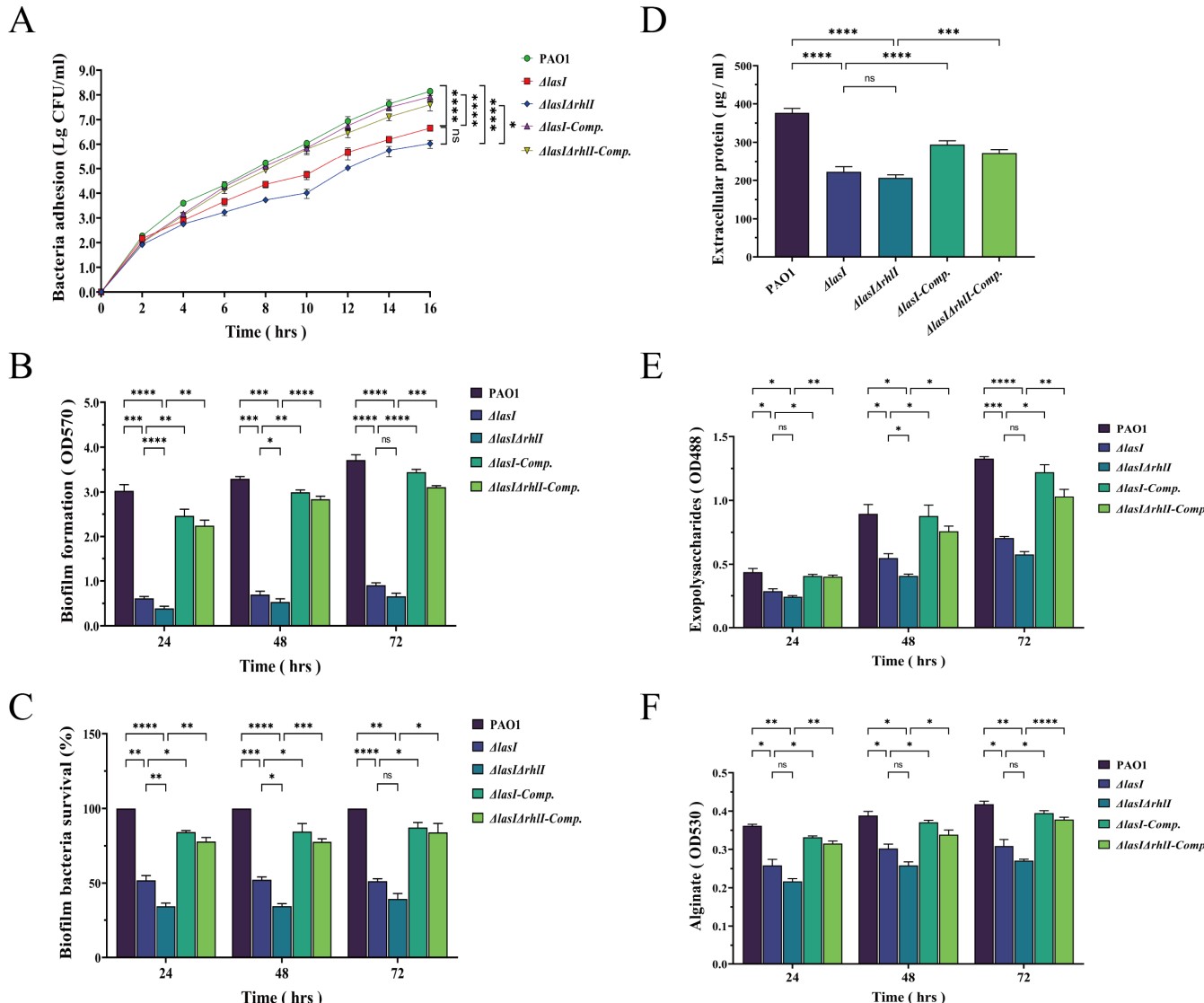

**FIG 1** The ability of adhesion, biofilm formation, and EPS of PAO1, *ΔlasI*, *ΔlasIΔrhlI*, *ΔlasI-Comp.*, and *ΔlasIΔrhlI-Comp.*. (A) Sixteen-hour bacterial adhesion assay. (B) Crystal violet staining assay to detect the biofilm content. (C) XTT reduction method to detect bacterial survival in biofilm. (D) BCA assay to detect protein concentrations. (E) Phenol-sulphuric acid assay to detect the content of exopolysaccharide. (F) Carbazole-sulphuric acid assay to determine alginate. All the data are presented as the M ± SD of three independent experiments. *, *P* < 0.05; **, *P* < 0.01; ***, *P* < 0.001; ****, *P* < 0.0001; ns, no significance.

knockout (Fig. 1C) ($P < 0.05$). Compared with PAO1 and the complemented strains, $\Delta lasI$ exhibited a survival rate of 51.93% ± 2.62% at 24 h, whereas $\Delta lasI\Delta rhlI$ had only 34.3% ± 1.74%. At 48 h, $\Delta lasI$ showed a survival rate of 52.38% ± 1.54%, whereas $\Delta lasI\Delta rhlI$ had 34.36% ± 1.39%. By 72 h, the survival rate of $\Delta lasI$ was 51.41% ± 1.29%, whereas that of $\Delta lasI\Delta rhlI$ was only 39.17% ± 3.00%.

When observing biofilm morphology under an inverted fluorescence microscope, we found that both PAO1 and the complemented strains exhibited compact and dense three-dimensional structures with bright fluorescence intensity. In contrast, the biofilm of $\Delta lasI$ appeared to have a dispersed distribution with weakened bacterial aggregation, whereas $\Delta lasI\Delta rhlI$ displayed an even looser structure with scattered bacteria and minimal clumping. Quantification of the mean fluorescence intensity revealed a consistent change tendency (Fig. S5A and B). A similar scenario was observed in scanning electron microscope (SME) imaging. The biofilm formed by PAO1 and the complemented strains exhibited densely packed cell clusters and abundant extracellular matrix coverage on tinfoil, whereas both the bacterial population and extracellular matrix within the biofilm showed a significant decrease from 24 h to 72 h for $\Delta lasI$ and $\Delta lasI\Delta rhlI$, particularly for $\Delta lasI\Delta rhlI$ (Fig. S5C).

## Mutation of *lasI*/*rhlI* reduces the EPS content of *P. aeruginosa*

Biofilm is a complex community of microorganisms composed of cellular biomass and EPS (26). EPS facilitates the adaptation of *P. aeruginosa* to its environment, with proteins and polysaccharides constituting 70%–80% of its total mass (27). The protein concentrations in the bacterial culture supernatants showed a significant decrease in $\Delta lasI$ and $\Delta lasI\Delta rhlI$ compared with PAO1 and the complemented strains (Fig. 1D) ($P < 0.05$). Visualization of the secreted proteins revealed notable differences among PAO1, $\Delta lasI$, $\Delta lasI\Delta rhlI$, $\Delta lasI$-*Comp.*, and $\Delta lasI\Delta rhlI$-*Comp.*. Particularly noteworthy was a protein band at approximately 33 kDa, which is consistent with the molecular weight of LasB elastase, a major virulence protein secreted by *P. aeruginosa* (28). In contrast, it was notably absent from the supernatants of $\Delta lasI$ and $\Delta lasI\Delta rhlI$ (Fig. S5D).

By visualizing the exopolysaccharides, it was observed that PAO1 colonies and their surroundings produced a distinct pink, whereas $\Delta lasI$ appeared slightly lighter pink and $\Delta lasI\Delta rhlI$ showed minimal to no coloration. However, $\Delta lasI$-*Comp.* and $\Delta lasI\Delta rhlI$-*Comp.* appeared even more vibrant pink (Fig. S5E). The quantitative results revealed a gradual increase in the secretion of exopolysaccharides for PAO1 over a period of 3 days. In contrast, the mutant strains consistently maintained lower levels, and the complemented strains were significantly reinstated (Fig. 1E) ($P < 0.05$). Alginate, as the major component of exopolysaccharides, exhibited a consistent increase in PAO1 and complemented strains, whereas $\Delta lasI$ and $\Delta lasI\Delta rhlI$ maintained low levels throughout the experiment (Fig. 1F) ($P < 0.05$).

## Mutation of *lasI*/*rhlI* attenuates the bacterial motility of *P. aeruginosa*

Bacterial motility plays a pivotal role in the initial stages of infection, facilitating colonization and subsequent biofilm formation. The motility of *P. aeruginosa* is primarily attributed to its flagella and pilus (29). Flagella facilitate swimming and swarming for individual and collective movement, respectively. By observing and measuring the swimming phenotypes on semi-solid agar surfaces, we discovered that $\Delta lasI$ and $\Delta lasI\Delta rhlI$ exhibited significantly diminished swimming distances compared with PAO1 and the complemented strains (Fig. S6A and B; Table S5) ($P < 0.05$). Regarding swarming, PAO1, $\Delta lasI$-*Comp.*, and $\Delta lasI\Delta rhlI$-*Comp.* displayed a larger rounded petal-like phenotype, which was not observed in $\Delta lasI$ and $\Delta lasI\Delta rhlI$ (Fig. S6C and D; Table S5) ($P < 0.05$). The distances of pilus-mediated twitching were also significantly reduced for $\Delta lasI$ and $\Delta lasI\Delta rhlI$ (Fig. S6E and F; Table S5) ($P < 0.05$). Furthermore, we quantified the instantaneous velocity, average speed, and average displacement on the X-axis relative to the initial position by capturing images of the bacteria's trajectories in the liquid medium. The

findings showed that the mutant strains resulted in a significantly reduced movement speed and shorter trajectories, whereas the complemented strains displayed faster and more linear trajectories (Fig. 2A through D; Videos S1 to S5; Table S5) ($P < 0.05$).

## Mutation of *lasI*/*rhlI* inhibits the virulence factors secretion of *P. aeruginosa*

LasA protease, a staphylococcal endopeptidase secreted by *P. aeruginosa* can specifically cleave the glycine pentapeptide cross-bridge in *S. aureus* peptidoglycan, leading to rapid lysis of *S. aureus* (20). The absorbance of heat-killed *S. aureus* treated with culture supernatants of PAO1, *ΔlasI-Comp.,* and *ΔlasIΔrhlI-Comp.* decreased over time, indicating strong LasA lysis activity in the supernatants. However, there was no significant decrease in turbidity when treated with *ΔlasI* and *ΔlasIΔrhlI* culture supernatants (Fig. 2E). By calculating the lysis rate at 150 min, we found that PAO1, *ΔlasI-Comp.,* and *ΔlasIΔrhlI-Comp.* culture supernatants exhibited significantly higher ability to lyse *S. aureus* (84.62% ± 3.79%, 79.90% ± 0.36%, and 75.28% ± 1.63%, respectively), whereas *ΔlasI* and *ΔlasIΔrhlI* culture supernatants could only lyse 12.43% ± 0.70% and 7.38% ± 0.41% (Fig. S6G; Table S5) ($P < 0.05$).

LasB elastase is the most abundant extracellular protease secreted by *P. aeruginosa*, responsible for degrading elastin (30). Both fluorescently labeled elastin and elastin Congo red assays showed significant elastin degradation activity in PAO1, whereas it was reduced in *ΔlasI* and almost absent in *ΔlasIΔrhlI* (Fig. 2F; Fig. S6H; Table S5) ($P < 0.05$). However, the complemented strains showed significantly restored LasB activity, consistent with the levels of *lasB* gene expression (Fig. S4D) and the observed protein bands (Fig. S5D).

Pyocyanin is a phenazine compound with reversible REDOX activity secreted by *P. aeruginosa* (31–33). Monitoring pyocyanin production for five consecutive days revealed a gradual increase in the green pigment of PAO1 over time (Fig. 2G). Compared with PAO1, the *ΔlasI* strain showed reduced but still significant production due to the predominant role of the Rhl system in pyocyanin biosynthesis. However, its content gradually decreased after 48 h, possibly due to the cascade effects of the Las system in the QS system and the partial degradation caused by pyocyanin's strong oxidative properties. Furthermore, knocking out both *lasI* and *rhlI* genes resulted in minimal pyocyanin production. Figure S6I showed the pigmentation on the PDP agar plate after 5 days of culture, showing markedly increased production of pyocyanin in *ΔlasI-Comp.* and *ΔlasIΔrhlI-Comp.*.

## Mutation of *lasI*/*rhlI* attenuates the cytotoxicity of THP-1 macrophages induced by *P. aeruginosa*

Due to the phagocytic capacity of macrophages, intracellular bacterial survival assays were conducted to evaluate the clearance of *P. aeruginosa* by THP-1 macrophages (Fig. 3A). The *ΔlasI* and *ΔlasIΔrhlI* infection groups exhibited a significant reduction in viable bacteria counts compared with the PAO1 infection group, suggesting substantial bacterial phagocytosis and more efficient macrophage-mediated clearance ($P < 0.05$). Moreover, there was a notable increase in viable counts of *ΔlasI-Comp.* and *ΔlasIΔrhlI-Comp.* ($P < 0.05$).

When evaluating the cytotoxicity induced by PAO1, *ΔlasI*, *ΔlasIΔrhlI*, *ΔlasI-Comp.,* and *ΔlasIΔrhlI-Comp.* culture supernatants, a significant reduction in cell viability was observed for all test groups compared with the control group, with a progressive decrease over time (Fig. 3B) ($P < 0.05$). In comparison to the PAO1 treated group, the cell viability of *ΔlasI*, and *ΔlasIΔrhlI* tested groups increased efficiently from 60.57% ± 4.67% to 77.29% ± 2.82% and 87.58% ± 5.46% at 6 h, from 44.68% ± 5.99% to 67.61% ± 2.54%, and 76.55% ± 4.38% at 12 h, as well as from 24.73% ± 2.17% to 42.42% ± 3.08% and 70.45% ± 2.29% at 24 h ($P < 0.05$).

Following 24 h treatment with bacterial culture supernatants, macrophages in the control group exhibited intact morphology with prominent antennas, a good refractive index, and an absence of particles. In the PAO1-treated group, the cells showed

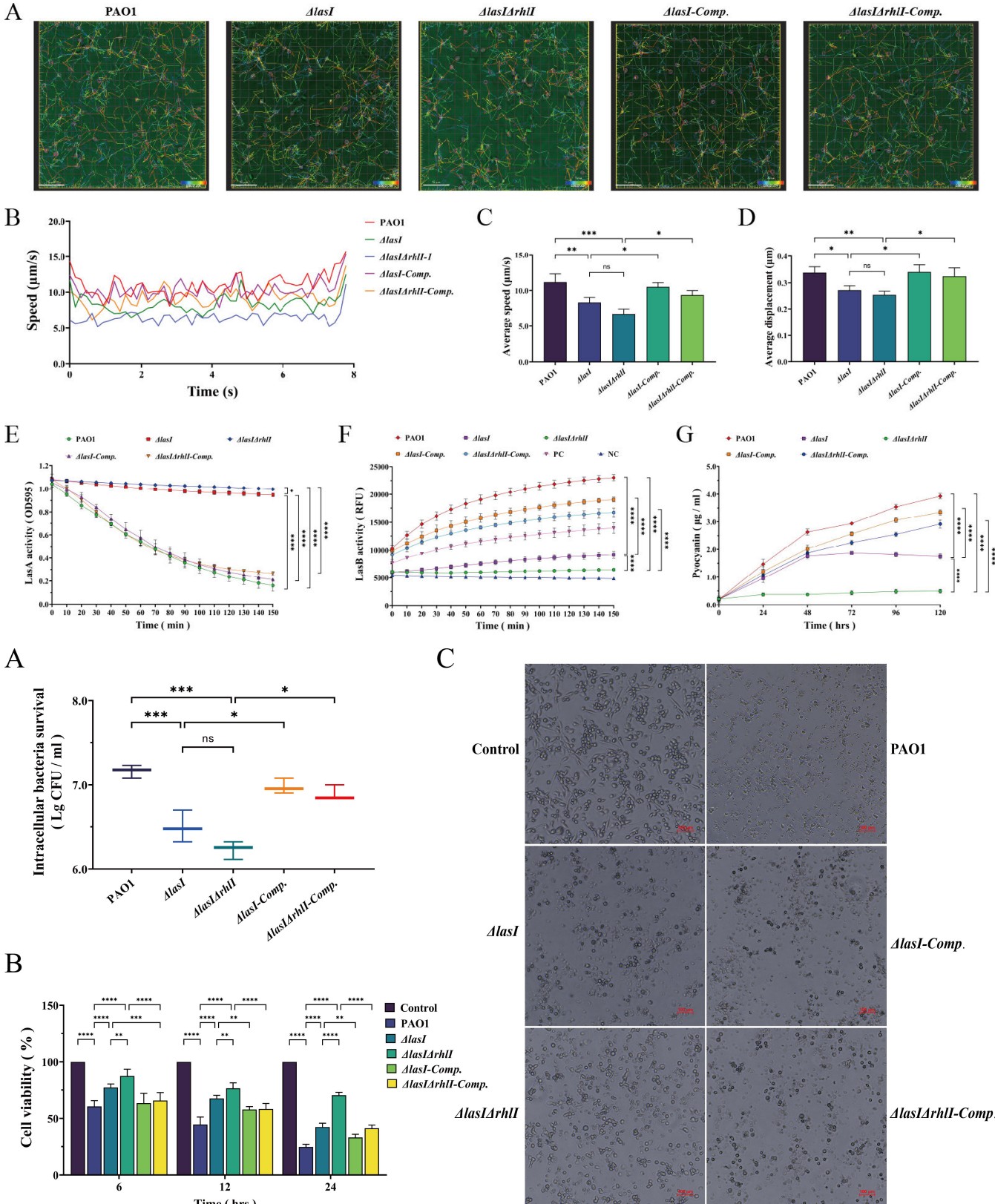

**FIG 3** Cytotoxic damage to THP-1 macrophages induced by PAO1, *ΔlasI*, *ΔlasIΔrhlI*, *ΔlasI-Comp.*, and *ΔlasIΔrhlI-Comp.*. (A) Intracellular bacteria survival when infecting THP-1 macrophages. (B) CCK-8 assay for cell viability of THP-1 macrophages treated with bacterial culture supernatants for 12 h, 24 h, and 48 h. (C) Cell morphology of THP-1 macrophages treated with bacterial culture supernatants for 24 h. Scale bar, 100 µm. All the data are presented as the M ± SD of three independent experiments. *, $P < 0.05$; **, $P < 0.01$; ***, $P < 0.001$; ****, $P < 0.0001$; ns, no significance.

Fig 2 (Continued)

**FIG 2** The motility ability and virulence factors of PAO1, *ΔlasI*, *ΔlasIΔrhlI*, *ΔlasI-Comp.*, and *ΔlasIΔrhlI-Comp..* (A) The motility tracking, (B) instantaneous velocity, (C) average speed of PAO1, and (D) average displacement on the X-axis relative to the initial position of *ΔlasI*, *ΔlasIΔrhlI*, *ΔlasI-Comp.*, and *ΔlasIΔrhlI-Comp.* in LB liquid medium. Scale bar, 15 µm. (E) LasA protease activity assay: lysis of *S. aureus* ATCC 29213 for 150 min. (F) LasB elastase activity assay: degradation of elastin fluorescent substrate for 150 min. (G) Production of pyocyanin for five consecutive days. All the data are presented as the M ± SD of three independent experiments. *, $P < 0.05$; **, $P < 0.01$; ***, $P < 0.001$; ****, $P < 0.0001$; ns, no significance.

noticeable damage characterized by blurred edges and increased granularity. The *ΔlasI*-treated group displayed slightly clearer cell edges and decreased granularity. In the *ΔlasIΔrhlI*-treated group, the cells appeared to have enhanced structural integrity with reduced granularity and visible tentacles in some cells. However, there was a significant decrease in cell viability and a noticeable disruption of cell integrity following treatment with the culture supernatants of complemented strains (Fig. 3C).

## Mutation of *lasI/rhlI* alleviates the oxidative stress of THP-1 macrophages induced by *P. aeruginosa*

ROS serves as a major mediator of cellular oxidative stress. Employing fluorescence microscopy for intracellular ROS observation, we noted a significant elevation in the fluorescence intensity of cells treated with PAO1, *ΔlasI-Comp.*, and *ΔlasIΔrhlI-Comp.* culture supernatants compared with the control group. In contrast, *ΔlasI* and *ΔlasIΔrhlI* exhibited diminished fluorescence intensity relative to PAO1 (Fig. 4A). Flow cytometry yielded similar results, showing significant differences in mean fluorescence intensity between the groups (Fig. 4B and C) ($P < 0.05$). Protein carbonylation is an indicator of protein oxidative damage. As depicted in Fig. 4D, treatment with PAO1 culture supernatant resulted in significantly increased protein carbonylation levels in THP-1 macrophages, whereas the deletion of *lasI* and *rhlI* led to a notable reduction in protein carbonylation content ($P < 0.05$). In addition, the protein and mRNA expression levels of the antioxidant enzymes SOD2 and GPX4 were significantly decreased following treatment with PAO1 culture supernatant. However, these levels showed a gradual increase upon deletion of *lasI* and *rhlI*, and decreased when *lasI* and *rhlI* were complemented (Fig. 4E through G; Tables S6 and S7) ($P < 0.05$). The findings suggest that PAO1 induces significant oxidative damage in macrophages and that mutation of *lasI* and *rhlI* within the QS system can alleviate this increased oxidative stress state.

## Mutation of *lasI/rhlI* alleviates the inflammatory response of THP-1 macrophages induced by *P. aeruginosa*

Exposure to external environmental stimuli can trigger the activation of inflammatory cytokines in macrophages, resulting in the initiation of inflammatory response (34). ROS is considered a critical determinant in the pathogenesis of inflammatory disorders. The protein and gene expression levels of inflammatory cytokines IL-1α, IL-1β, IL-6, IL-10, IL-12, and TNF-α were significantly upregulated in THP-1 macrophages treated with PAO1 compared with the control group. In comparison to the PAO1 treatment group, treatment with *ΔlasI* and *ΔlasIΔrhlI* resulted in a significant reduction in pro-inflammatory cytokines IL-1α, IL-1β, IL-6, IL-12, and TNF-α, along with an elevation in anti-inflammatory cytokine IL-10 (Fig. 5A through F; Tables S7 and S8) ($P < 0.05$). Simultaneously, there was a notable increase in the intracellular activity of inflammatory-inducing enzymes COX-2 and iNOS upon PAO1-induced stimulation, whereas *ΔlasI* and *ΔlasIΔrhlI* led to a substantial reduction. Conversely, when *lasI/rhlI* genes were complemented, there was a tendency for upregulation of pro-inflammatory cytokines and inflammatory-inducing enzymes, as well as downregulation of IL-10 (Fig. 5G through I; Tables S6 to S8) ($P < 0.05$). The NF-κB signaling pathway plays a pivotal role in inflammatory response. Our findings revealed that treatment with PAO1 and the complemented strains significantly upregulated the protein expression levels of Phospho-IκBα/IκBα and Phospho-NF-κB p65/NF-κB p65 compared with the control group. In contrast, there was a significant decrease in

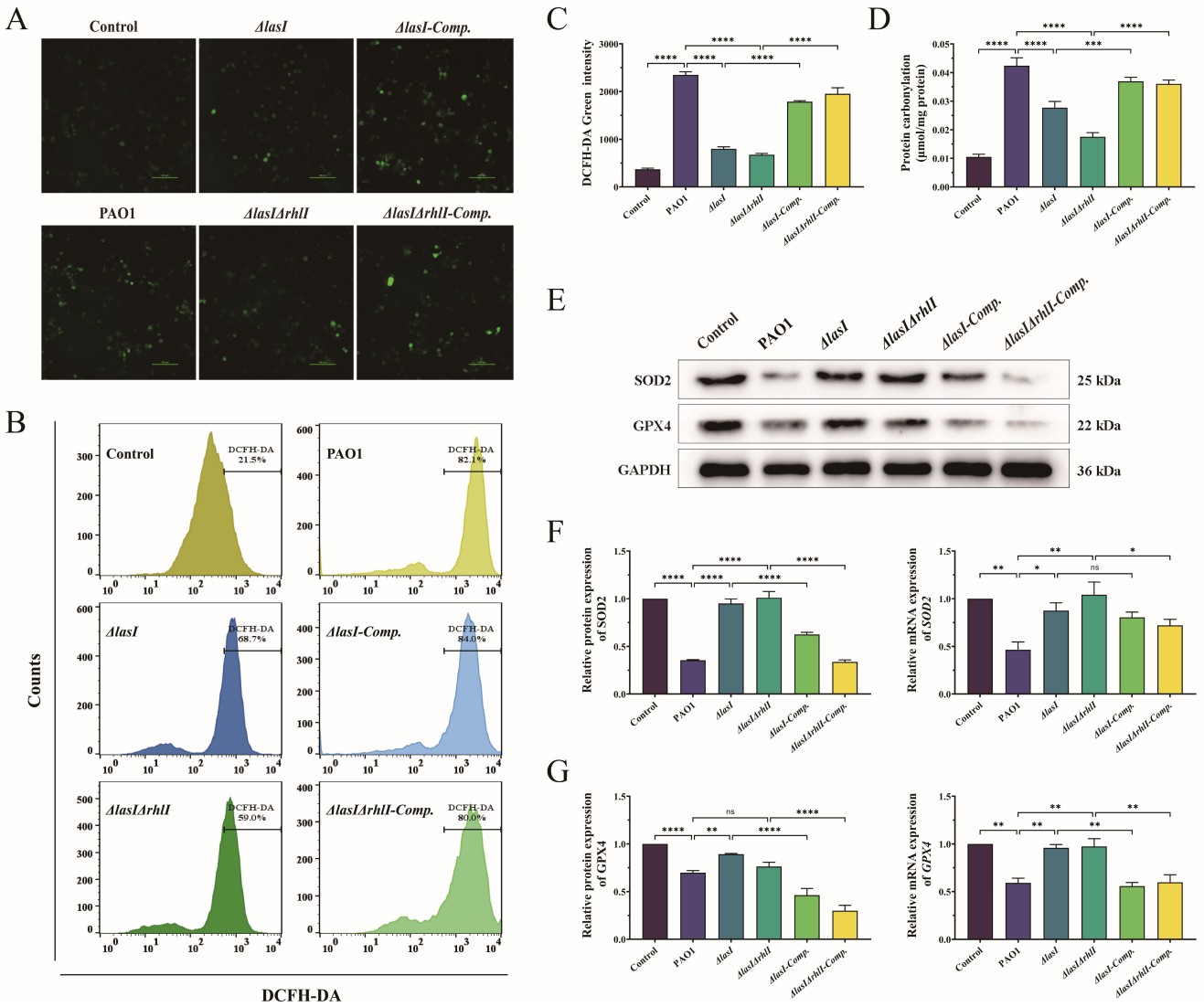

**FIG 4** Oxidative stress in THP-1 macrophages induced by PAO1, *ΔlasI*, *ΔlasIΔrhlI*, *ΔlasI-Comp.*, and *ΔlasIΔrhlI-Comp.*. (A) Intracellular ROS observed by inverted fluorescence microscopy. DCFH-DA fluorescent labeling. Scale bar, 100 µm. (B) Intracellular ROS detected by flow cytometry and (C) the mean fluorescence intensity. (D) Protein carbonyl content. (E) Western blot of SOD2 and GPX4. (F) Relative protein and mRNA expression level of SOD2. (G) Relative protein and mRNA expression level of GPX4. All the data are presented as the M ± SD of three independent experiments. *, $P < 0.05$; **, $P < 0.01$; ***, $P < 0.001$; ****, $P < 0.0001$; ns, no significance.

these levels following treatment with *ΔlasI* and *ΔlasIΔrhlI*. A similar tendency was observed for the mRNA expression levels of *IκBα* and *NF-κB p65* (Fig. 5J through L; Tables S6 and S7) ($P < 0.05$).

## Mutation of *lasI*/*rhlI* reduces DNA damage and apoptosis of THP-1 macrophages induced by *P. aeruginosa*

The induction of apoptosis is initiated by various factors, including cellular stress and DNA damage, leading to the upregulation of pro-apoptosis proteins and the inhibition of anti-apoptosis regulatory proteins (35). The TUNEL assay revealed a significant increase in DNA damage in THP-1 macrophages following treatment with PAO1 culture supernatant. However, a notable reduction was observed due to the mutation of *lasI* and *rhlI*, and there was a significant increase after complementation with *lasI* and *rhlI* (Fig. 6A). The flow cytometry analysis showed that treatment with PAO1 culture supernatant led to a

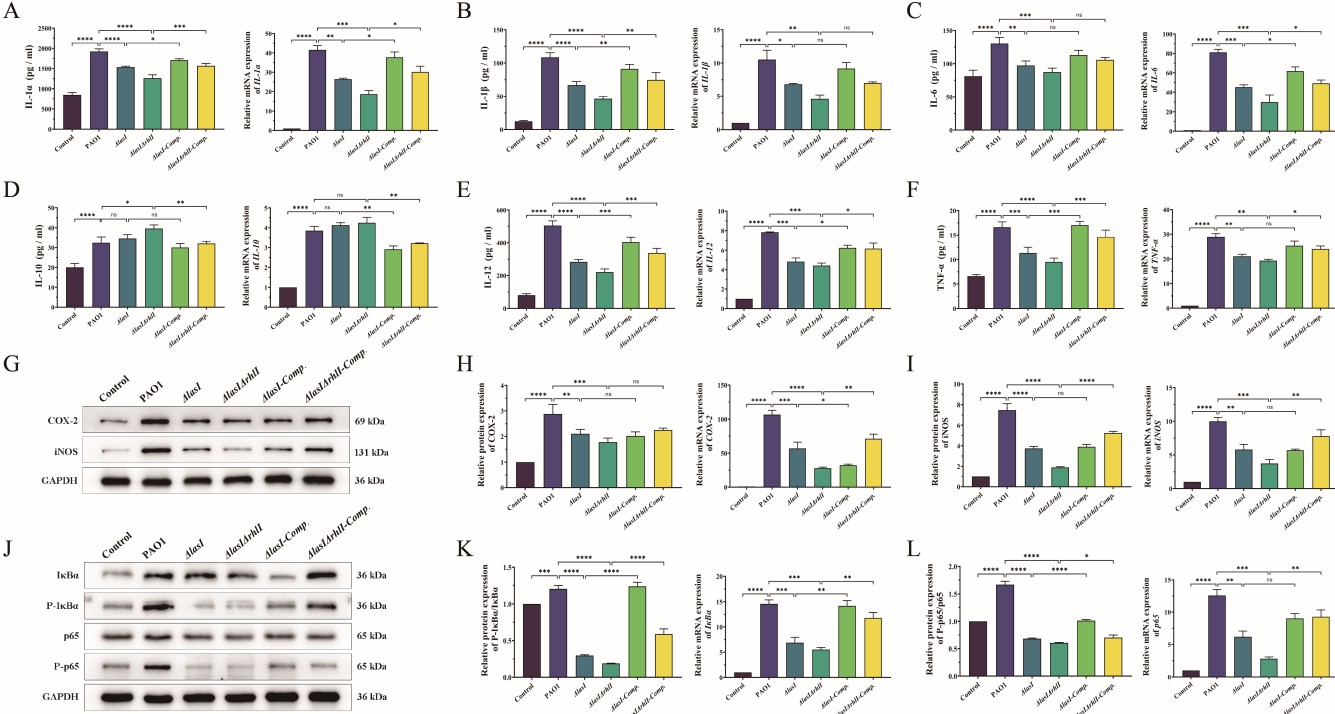

**FIG 5** The inflammatory response in THP-1 macrophages induced by PAO1, *ΔlasI*, *ΔlasIΔrhlI*, *ΔlasI-Comp.*, and *ΔlasIΔrhlI-Comp.*. The protein concentration and relative mRNA expression levels of (A) IL-1α, (B) IL-1β, (C) IL-6, (D) IL-10, (E) IL-12, and (F) TNF-α determined by ELISA and RT-qPCR. (G) Western blot of COX-2 and iNOS. (H) Relative protein and mRNA expression level of COX-2. (I) Relative protein and mRNA relative expression level of iNOS. (J) Western blot of IκBα, Phospho-IκBα, NF-κB p65, and Phospho-NF-κB p65. (K) Relative protein expression level of P-IκBα/ IκBα and relative mRNA expression level of *IκBα*. (L) Relative protein expression level of P-p65/p65 and relative mRNA expression level of *p65*. All the data are presented as the M ± SD of three independent experiments. *, $P < 0.05$; **, $P < 0.01$; ***, $P < 0.001$; ****, $P < 0.0001$; ns, no significance.

significant increase in the apoptosis rate of THP-1 macrophages, rising from 10.95% ± 0.47% in the control group to 51.98% ± 4.04% (including early and late apoptotic cells). Notably, the *ΔlasI* and *ΔlasIΔrhlI* treatments resulted in reduced apoptosis rates of 26.8% ± 2.74% and 20.15% ± 0.57%, respectively. However, the introduction of *ΔlasI-Comp.* and *ΔlasIΔrhlI-Comp.* restored the apoptosis rate to 43.87% ± 4.53% and 37.38% ± 5.11%, respectively (Fig. 6B and C) ($P < 0.05$). The western blot analysis further demonstrated that treatment with PAO1, *ΔlasI-Comp.*, and *ΔlasIΔrhlI-Comp.* culture supernatants resulted in prominent upregulation of the pro-apoptosis protein Bax and inhibition of the anti-apoptosis protein Bcl-2. Conversely, the mutation of *lasI* and *rhlI* led to a significant downregulation of Bax and upregulation of Bcl-2 (Fig. 6D through F; Tables S6 and S7) ($P < 0.05$). The balance between Bcl-2 and Bax determines the propensity of cellular apoptosis. Treatment with PAO1 led to a decrease in the Bcl-2/Bax ratio, promoting apoptosis, whereas treatment with *ΔlasI* and *ΔlasIΔrhlI* resulted in an elevation of the ratio, indicating inhibition of apoptosis. Additionally, *ΔlasI-Comp.* and *ΔlasIΔrhlI-Comp.* exhibited promotion of apoptosis (Fig. 6G; Table S6) ($P < 0.05$).

## DISCUSSION

*P. aeruginosa* is a gram-negative bacterium posing a significant threat to immunocompromised individuals and healthcare settings (36, 37). Its propensity for developing antibiotic resistance poses a significant challenge in the management of *P. aeruginosa* infection (38, 39). In recent years, anti-virulence therapy has emerged as a potentially revolutionary therapeutic approach to *P. aeruginosa* infection (40). Therefore, an in-depth understanding of the virulence and pathogenic mechanisms underlying the infection is critical for implementing this strategy.

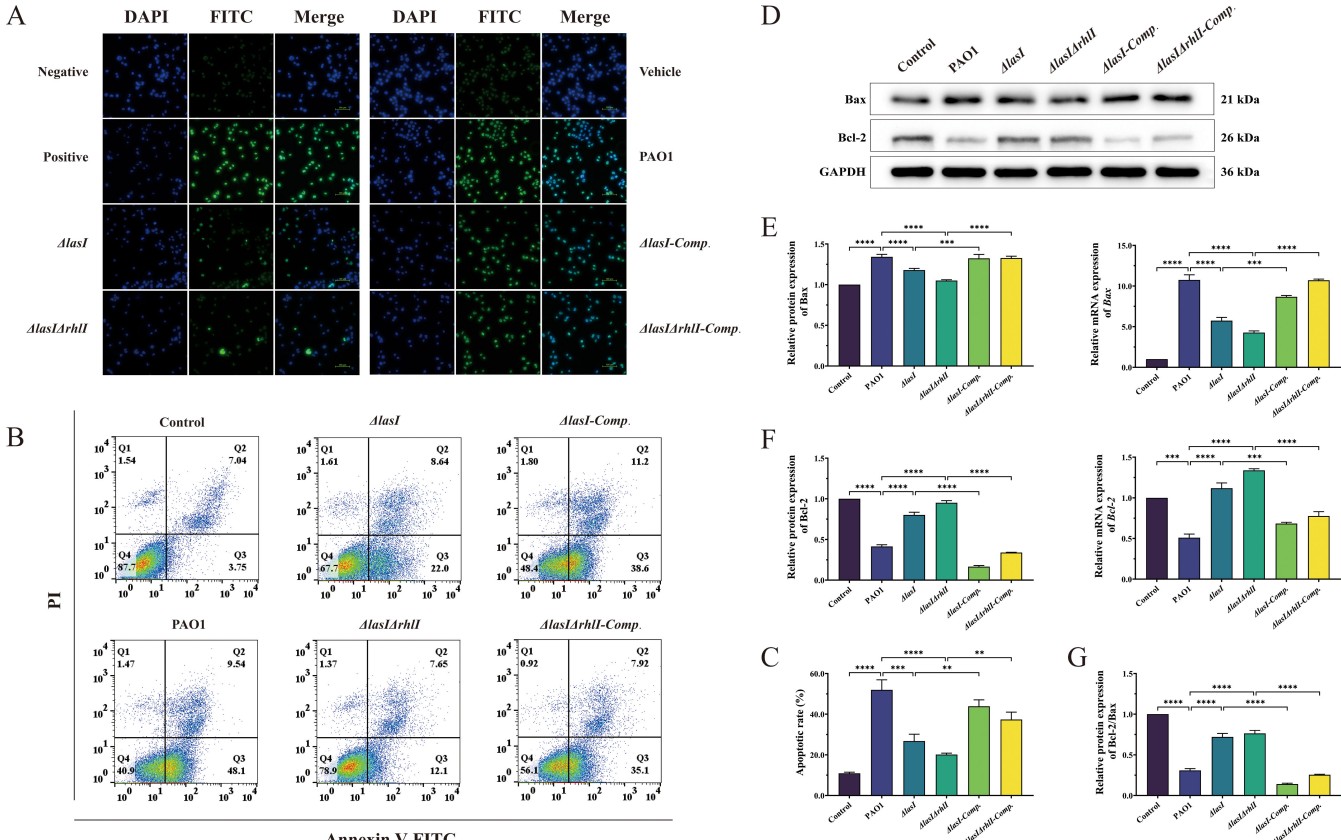

**FIG 6** The DNA damage and apoptosis in THP-1 macrophages induced by PAO1, *ΔlasI*, *ΔlasIΔrhlI*, *ΔlasI-Comp.*, and *ΔlasIΔrhlI-Comp.*. (A) DNA damage of THP-1 macrophages observed by inverted fluorescence microscopy. FITC and DAPI double fluorescent labeling. Scale bar, 100 µm. (B) Apoptosis of THP-1 macrophages detected by flow cytometry and (C) statistical analysis of apoptosis rates in three independent replicates. (D) Western blot of Bax and Bcl-2. (E) Relative protein and mRNA expression levels of Bax. (F) Relative protein and mRNA expression levels of Bcl-2. (G) Relative protein expression level of Bcl-2/Bax. All the data are presented as the M ± SD of three independent experiments. *, $P < 0.05$; **, $P < 0.01$; ***, $P < 0.001$; ****, $P < 0.0001$; ns, no significance.

The QS system of *P. aeruginosa* is responsible for regulating the production of virulence factors, biofilm formation, and motility behavior, which are critical in host-pathogen interactions (41). Studies have shown that the Las and Rhl systems intricately regulate approximately 10% of *P. aeruginosa* genes, encompassing diverse physiological processes and virulence traits (42). Thus, the mutation of upstream encoding genes *lasI* and *rhlI* in the QS system is anticipated to exert an impact on the virulence and pathogenicity of *P. aeruginosa*. In this study, the transcriptome sequencing and mRNA expression analysis revealed that differences between mutant strains and PAO1 were specifically focused on the key regulatory genes of the QS system and its downstream virulence-associated genes (Fig. S3 and S4; Table 1). This further confirms the role of *lasI* and *rhlI* as key regulatory promoters in the Las and Rhl systems, respectively, along with their cascading influence on the Pqs system.

*P. aeruginosa* has the capacity to develop biofilms on both biotic and abiotic surfaces, such as contact lenses and catheters (43). Once established, these biofilms facilitate bacterial evasion of host immunity and confer resistance to antibiotic clearance, enabling survival in adverse environments and leading to chronic and persistent infections (44, 45). The formation of biofilm is associated with the adhesion of planktonic bacteria (46, 47). Our research has demonstrated that the mutation of *lasI* and *rhlI* can effectively inhibit bacterial adhesion and biofilm formation directly, resulting in reduced biofilm biomass, impaired bacterial survival within the biofilms, and a looser biofilm structure (Fig. 1A through C; Fig. S5A through C). EPS is a major component of the biofilm matrix and plays a vital role in enhancing surface adhesion and intercellular

cohesion, thereby providing protection and structural integrity for mature biofilms (46, 48). Meanwhile, viscous alginate is involved in the mechanism of anti-phagocytosis and assists *P. aeruginosa* in resisting free radicals and antibiotics (26). In the investigation, a notable reduction in extracellular proteins and polysaccharides was observed in *ΔlasI* and *ΔlasIΔrhlI* strains, which contributed to the disruption of the microenvironment essential for biofilm formation (Fig. 1D through F; Fig. S5D and E).

The formation and development of biofilms are closely associated with bacterial motility, which contributes to the remarkable adaptability of *P. aeruginosa* in diverse environments (36, 49). *P. aeruginosa* employs multiple mechanisms for surface and fluid movement, including swimming, swarming, and twitching (50, 51). Swimming is an individual behavior, whereas swarming is a collective behavior driven by flagella. Twitching, on the other hand, is caused by the contraction and extension of type IV pili (TFP) (52–56). Bacterial motility plays a pivotal role in bacterial adhesion, biofilm formation, and host-pathogen interactions, synergistically influencing the migration and diffusion of *P. aeruginosa* (55, 57, 58). Knocking down *lasI* and *rhlI* of the QS system resulted in significantly reduced motility, as evidenced by decreased travel distance and covered area on semi-solid surfaces, as well as a reduction in bacterial average movement speed and displacement in liquid medium (Fig. S6A through F; Fig. 2A through D; Videos S1 to S5). These findings suggest that altered individual cell behavior drives swarming, twitching, and biofilm development.

The extracellular virulence factors LasA Protease, LasB elastase, and pyocyanin play a crucial role in the pathogenicity of *P. aeruginosa*. Specifically, the LasA protease, also known as *S. aureus* hemolysin, induces cell death by disrupting the host cell membrane and breaking down intercellular junctions, thereby compromising the host epithelial barrier and facilitating bacterial invasion into tissues (22, 59–61). LasB elastase, secreted via the Type II Secretion System (T2SS) (30, 51), is a pivotal virulence factor in *P. aeruginosa* infection by degrading tissue proteins, extracellular matrix components, and immune system components (24, 62). Numerous studies have emphasized its significant contribution to pathogenicity (63–65). Pyocyanin is a REDOX-active factor that participates in inhibiting cellular respiration, disrupting intracellular REDOX balance, inducing host damage, and contributing to immune evasion (33, 66, 67). In this investigation, the levels of LasA, LasB, and pyocyanin in the culture supernatants of *ΔlasI* and *ΔlasIΔrhlI* strains showed a significant reduction to exceedingly low levels when compared with PAO1, whereas the complemented strains recovered obviously (Fig. 2E through G; Fig. S6G through I). These findings imply that deleting key regulatory genes of the QS system could be beneficial in attenuating bacterial proliferation and dissemination within the host, as well as reducing damage to both the host's immune system and REDOX system. In summary, the above results illustrate that the mutation of *lasI/rhlI* genes in the QS system can greatly decrease the virulence of *P. aeruginosa*.

Macrophages are integral in the phagocytosis of invading pathogens, serving as the vanguard of the innate immunity system (11, 12, 68). The intracellular killing assay indicates that the mutant strains are more susceptible to being engulfed and scavenged by THP-1 macrophages compared with PAO1, indirectly demonstrating that their attenuated virulence is conducive to host immune clearance (69, 70). Furthermore, the cell viability and cell morphology of THP-1 macrophages further illustrate that the mutant strains with attenuated virulence contribute to a reduction in cytotoxicity toward macrophages induced by *P. aeruginosa* (Fig. 3).

When the production of ROS exceeds the buffering capacity of the antioxidant defense system, resulting in an imbalance of the REDOX state, oxidative stress occurs (71). This leads to DNA hydroxylation, protein denaturation, tissue damage, and ultimately cell lysis and death through apoptosis, necrosis, and autophagy (72). Our findings demonstrate that the extracellular secretory components of PAO1 induce excessive production of ROS, leading to the rapid demise of activated macrophages, which aligns with previous assertions (73, 74). The protein carbonylation, which is a non-reversible chemical modification indicating protein oxidation damage, was also

observed in THP-1 macrophages following exposure to PAO1 culture supernatant. To protect cells from oxidative damage mediated by ROS free radicals, the antioxidant defense system synthesizes antioxidant enzymes such as superoxide dismutase (SOD), catalase (CAT), and glutathione peroxidase (GPX) (75). In our research, the culture supernatant of PAO1 significantly affected the cellular antioxidant system, resulting in reduced activity of antioxidant enzymes (SOD2 and GPX4) and impaired ability to scavenge ROS, thereby exacerbating cell damage and ultimately leading to cell death. However, the deletion of *lasI* and *rhlI* genes in the QS system not only resulted in a substantial decrease in ROS levels and protein carbonylation but also led to a significant increase in antioxidant enzymes, thereby maintaining cellular REDOX homeostasis and preventing ROS-induced oxidative damage (Fig. 4; Tables S6 and S7). Consequently, inhibiting the QS system holds promise in protecting THP-1 macrophages against oxidative stress induced by *P. aeruginosa*.

Inflammation can be triggered by oxidative stress. When invaded by pathogens or other foreign bodies, elevated levels of ROS contribute to the excessive activation of inflammatory cytokines in macrophages, thus triggering an inflammatory response (34, 76, 77). In our study, the NF-κB signaling pathway was activated, and inflammatory

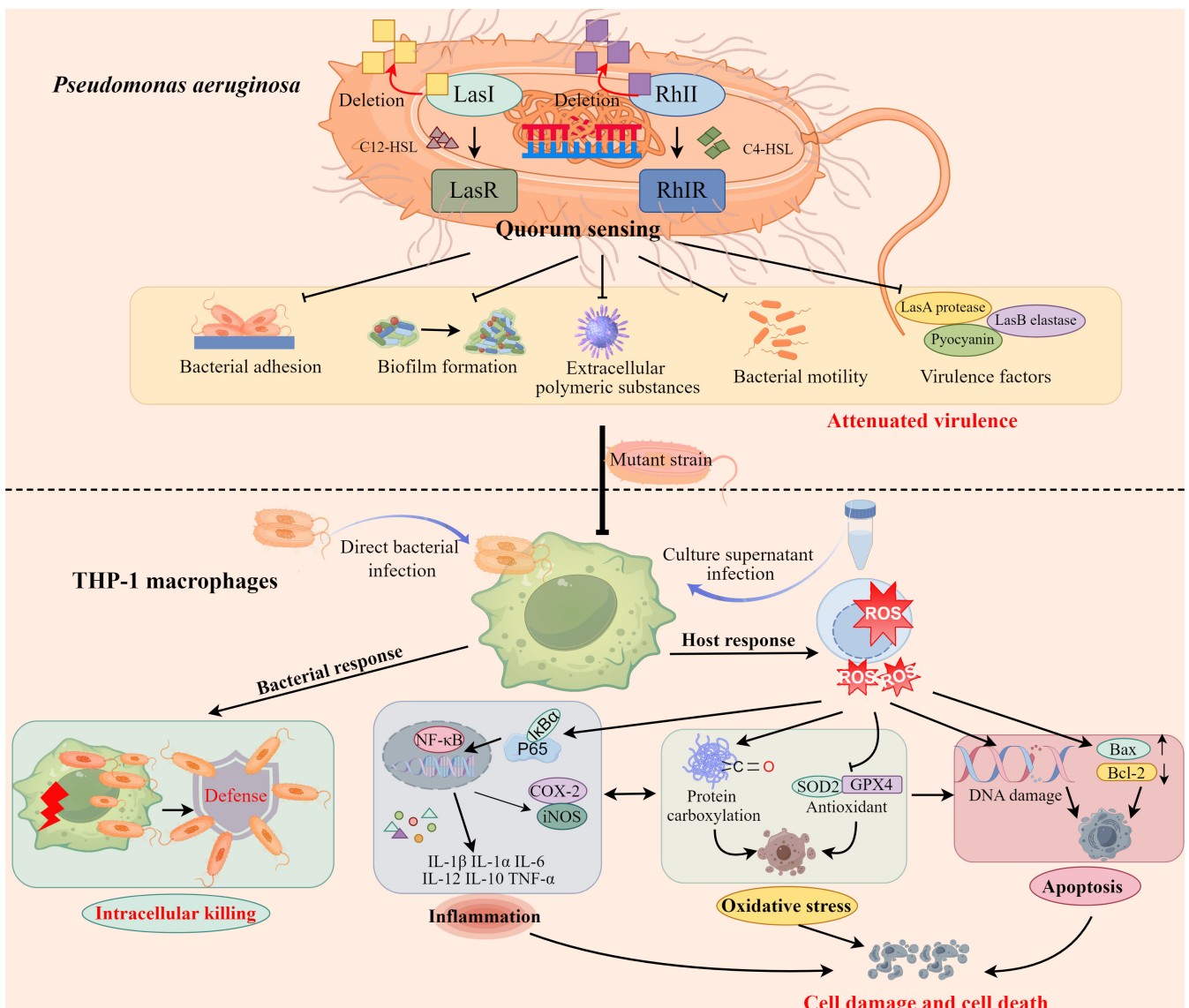

**FIG 7** Effect of *lasI/rhlI* gene deletion in the QS system of *P. aeruginosa* on THP-1 macrophages.

cytokines IL-1α, IL-1β, IL-6, IL-10, IL-12, and TNF-α were released in THP-1 macrophages following treatment with PAO1 culture supernatant. Additionally, the inflammation-inducing enzymes COX-2 and iNOS were also induced in response to inflammation, facilitating macrophages' involvement in immune responses. However, excessive iNOS leads to the consumption of arginine and the generation of a high level of peroxide-free radical NO. Consequently, this causes cellular metabolic hypoxia and disrupts mitochondrial oxidative phosphorylation function, ultimately resulting in the release of substantial amounts of ROS (78). Significantly reduced levels of pro-inflammatory cytokines and inflammation-inducing enzymes were observed following treatment with Δ*lasI* and Δ*lasI*Δ*rhlI*, accompanied by an elevated trend for the anti-inflammatory cytokine IL-10. Meanwhile, the NF-κB inflammatory pathway was inhibited by the mutants (Fig. 5; Tables S6 to S8). These findings suggest that the mutation of *lasI*/*rhlI* in the QS system may attenuate the inflammatory response induced by *P. aeruginosa* in THP-1 macrophages through the inactivation of NF-κB signaling pathway.

In our research, an excessive accumulation of ROS was observed to enter the nucleus through the cytoplasm and nuclear pores when exposed to PAO1 culture supernatant. This resulted in DNA damage and apoptosis in THP-1 macrophages, accompanied by a significant upregulation of the pro-apoptotic protein Bax and downregulation of the anti-apoptotic protein Bcl-2. When the equilibrium between the two apoptotic proteins is disrupted, it intensifies the apoptotic process. However, this detrimental effect can be effectively reversed by deleting the *lasI* and *rhlI* genes, thereby reducing the DNA damage and apoptosis of THP-1 macrophages induced by the extracellular virulence components of *P. aeruginosa*, thus preventing rapid damage and death of THP-1 macrophages (Fig. 6; Tables S6 and S7). These findings indicate that the mutation of *lasI*/*rhlI* genes in the QS system can inhibit the DNA damage and apoptosis of THP-1 macrophages induced by *P. aeruginosa*.

In conclusion, our work demonstrates that mutation of *lasI*/*rhlI* genes in the QS system exerts detrimental effects on bacterial adhesion, biofilm formation, EPS generation, bacterial motility, and virulence factors production of *P. aeruginosa*. Importantly, concomitant with attenuated virulence, the cytotoxicity, oxidative stress, inflammation, and apoptosis induced by *P. aeruginosa* in THP-1 macrophages are also significantly alleviated (Fig. 7). These findings comprehensively illustrate the critical role of the QS system in *P. aeruginosa* pathogenesis and imply that intervening in the QS system of *P. aeruginosa* is anticipated to mitigate its pathogenicity against macrophages.

## ACKNOWLEDGMENTS

This work was kindly supported by the Henan Province Chinese Medicine Scientific Research Special Project (grant number: 2022ZY1078; 2023ZY1011) and the Key Research Projects of Henan Higher Education Institutions (grant number: 22A360008).

Y.R. conducted most of the experiments and wrote the final version of the manuscript. X.L. and Y.L. conceived and designed the study. X.Y. and R.Z. helped to draft the manuscript. D.L., C.W., and Z.H. conducted part of the experiments and data analysis. Y.H. and Y.L. participated in collecting data. All authors reviewed and approved the final manuscript.

## AUTHOR AFFILIATIONS

[1]Dazhou integrated Traditional Chinese Medicine & Western Medicine Hospital, Dazhou Second People's Hospital, Dazhou, China
[2]Henan University of Chinese Medicine, Zhengzhou, China
[3]Henan Province Hospital of Traditional Chinese Medicine, The Second Affiliated Hospital of Henan University of Chinese Medicine, Zhenghzhou, China
[4]The Key Laboratory of Pathogenic Microbes &Antimicrobial Resistance Surveillance of Zhengzhou, Zhengzhou, China

[5]Henan Engineering Research Center for Identification of Pathogenic Microbes, Zhengzhou, China
[6]Henan Provincial Key Laboratory of Antibiotics-Resistant Bacterial Infection Prevention & Therapy with Traditional Chinese Medicine, Zhengzhou, China

## AUTHOR ORCIDs

Xinwei Liu  http://orcid.org/0000-0002-5358-886X
Yongwei Li  http://orcid.org/0000-0003-2297-8934

## FUNDING

| Funder | Grant(s) | Author(s) |
| --- | --- | --- |
| Health Commission of Henan Province (河南省卫生健康委员会) | 2022ZY1078 | Yongwei Li |
| Health Commission of Henan Province (河南省卫生健康委员会) | 2023XY1011 | Yongwei Li |
| Education Department of Henan Province (河南省教育厅) | 22A360008 | Xinwei Liu |

## AUTHOR CONTRIBUTIONS

Yanying Ren, Data curation, Formal analysis, Investigation, Methodology, Project administration, Resources, Software, Supervision, Validation, Visualization, Writing – original draft, Writing – review and editing | Xiaojuan You, Data curation, Formal analysis, Writing – original draft, Writing – review and editing | Rui Zhu, Formal analysis, Methodology, Writing – original draft, Writing – review and editing | Dengzhou Li, Conceptualization, Investigation, Methodology, Writing – original draft, Writing – review and editing | Chunxia Wang, Conceptualization, Methodology, Supervision, Writing – original draft, Writing – review and editing | Zhiqiang He, Data curation, Formal analysis, Methodology, Writing – original draft, Writing – review and editing | Yue Hu, Investigation, Methodology, Writing – original draft, Writing – review and editing | Yifan Li, Data curation, Formal analysis, Investigation, Methodology, Validation, Visualization, Writing – original draft, Writing – review and editing | Xinwei Liu, Conceptualization, Writing – original draft, Writing – review and editing | Yongwei Li, Conceptualization, Funding acquisition, Investigation, Project administration, Resources, Supervision, Validation, Writing – original draft, Writing – review and editing

## DATA AVAILABILITY

All data generated or analyzed during this study are included in the submitted manuscript.

## ETHICS APPROVAL

This study does not contain any experiments with human participants or animals performed by any of the authors.

## ADDITIONAL FILES

The following material is available online.

### Supplemental Material

**Supplementary material (Spectrum04146-23-s0001.docx).** Fig. S1 to S6; Videos S1 to S5; Tables S1 to S8.
**Video S1 (Spectrum04146-23-s0002.mp4).** Tracking the 2D trajectory of PAO1 in LB liquid medium.

**Video S2 (Spectrum04146-23-s0003.mp4).** Tracking the 2D trajectory of *ΔlasI* in LB liquid medium.

**Video S3 (Spectrum04146-23-s0004.mp4).** Tracking the 2D trajectory of *ΔlasI ΔrhlI* in LB liquid medium.

**Video S4 (Spectrum04146-23-s0005.mp4).** Tracking the 2D trajectory of *ΔlasI*-Comp in LB liquid medium.

**Video S5 (Spectrum04146-23-s0006.mp4).** Tracking the 2D trajectory of *ΔlasI ΔrhlI*-Comp in LB liquid medium.

## Open Peer Review

**PEER REVIEW HISTORY (review-history.pdf).** An accounting of the reviewer comments and feedback.

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
