## [Reviewer comments · Microbiology Spectrum]

Microbiology Spectrum

Mutation of *Pseudomonas aeruginosa* lasI/rhlI diminishes its cytotoxicity, oxidative stress, inflammation and apoptosis on THP-1 macrophages

Yanying Ren, Xiaojuan You, Rui Zhu, Dengzhou Li, Chunxia Wang, Zhiqiang He, Yue Hu, Yifan Li, Xinwei Liu, and Yongwei Li

Corresponding Author(s): Xinwei Liu, Henan Province Hospital of TCM

Review Timeline:

Submission Date:	December 7, 2023
Editorial Decision:	January 8, 2024
Revision Received:	March 7, 2024
Editorial Decision:	March 27, 2024
Revision Received:	May 24, 2024
Editorial Decision:	June 7, 2024
Revision Received:	June 19, 2024
Accepted:	June 27, 2024

Editor: Dhammika Navarathna

Reviewer(s): The reviewers have opted to remain anonymous.

Transaction Report:

DOI: <https://doi.org/10.1128/spectrum.04146-23>

Re: Spectrum04146-23 (Mutation of *Pseudomonas aeruginosa* lasI/rhlI diminishes its cytotoxicity, inflammatory response and oxidative stress on THP-1 macrophages)

Dear Dr. Xinwei Liu:

Thank you for the privilege of reviewing your work. Below you will find my comments, instructions from the Spectrum editorial office, and the reviewer comments.

Revision Guidelines

Sincerely,
Dharmika Navarathna
Editor
Microbiology Spectrum

Reviewer #1 (Comments for the Author):

Ren et al. report on findings that the lasI and rhlI genes are involved in pathogenic behaviors of the widespread bacterial pathogen *Pseudomonas aeruginosa*. This research manuscript described data using an array of assays to investigate the behavior of mutants lacking one or both of these genes relative to that of the WT. The figures are well put together and the data support most of the author's conclusion. The primary thing lacking in this work is an organized narrative that synthesizes what is

new about this work. For this reason, I recommend a concluding figure with a model of their system of interest that depicts how the author's data have expanded understanding of quorum-sensing in PA pathogenesis. The author's major concern is that throughout the text (for example Line 84-90) the authors vastly overstate the importance of their findings. Their assays are in cell culture, and they do not directly test virulence or infection. The authors extrapolate from their in vitro studies that biofilm formation, and some other bacterial systems, play a "prominent role in attenuating virulence." Moreover, these in vitro assays were performed under normoxic conditions with no control for atmosphere, despite the hypoxic conditions in which PA infections naturally experience in the lung and soft tissue infections. Thus, these claims (such as "targeting the quorum sensing system may be an ideal option for infection control and treatment") must be softened and brought into line with their evidence.

Other concerns are below.

The authors do not employ the use of complement strains, in which *lasI*, *rhlI* are added back. Thus, they cannot exclude the possibility that some of the phenotypes they observe are either from additional culturing and domestication of the mutant strains. This may be relevant for some of the milder phenotypes, for instance the bacteria adhesion assay timecourse, which shows little difference between the strains (5A).

Line 59. The authors should explain and put into context what quorum sensing is, and why it plays a role in pathogenesis.

Line 63. Similarly, the authors must explain why these mutants relate to quorum sensing.

Line 63. The relevance of THP-1 macrophages to *P. aeruginosa* pathogenesis is not explained.

Line 309-316. Measuring Congo Red dye is not a measure of c-di-GMP, but a measure of EPS.

Line 438-466 (and elsewhere): Keep methods in the methods section. Do not describe in detail bacterial mutant generation in the results. This also does not require a main text figure (1 & 2), though that could be included in the supplemental.

Line 541-546: These microscopy analyses should be quantified, with multiple replicates, to justify these claims. You cannot draw this conclusion from single images alone.

Fig. 6B: You should denote the band of interest with a marking on the image.

Line 576. Incorrect. You did not visualize c-di-GMP levels by staining for EPS. You infer c-di-GMP levels from this assay. Please revise to be correct.

Line 582-594: These are rather archaic methods to analyze bacterial motility. I recommend using microscopy and live imaging of these behaviors, as is standard practice now in the study of bacterial motility. Further, the type of motility being observed is often incorrectly attributed through these types of plate/agar-based assays.

Line 616: What is a 'redox ability' ?

Line 631: Why did you select to work with macrophages? My understanding is that it is neutrophils that play a more dominant role in clearing PA infection (<https://www.ncbi.nlm.nih.gov/pmc/articles/PMC5371898/>). These cells have very different ROS-generation behavior and abilities (NETosis, for example).

Line 683 and Fig. 11: Without quantitative data on ROS concentrations you cannot say whether or not "PA01 induces a significant oxygen burst in macrophages...can alleviate this increased oxidative stress state." ROS have healthy normal levels that are part of normal eukaryotic cell processes. Most cells, bacterial and eukaryotic, are also exceptionally well-equipped to eliminate even high levels of ROS through catalase and peroxiredoxin enzymes. Hence, 'oxidative stress' needs to be proven by a method such as quantifying protein carbonylation or DNA damage. The authors have not done so, and should limit their claims to relative increases or decreases of fluorescence. Moreover, it isn't clear there is a large difference in the images presented in Fig. 11A. Can the authors confirm that the images shown were captured with identical background and thresholding levels?

Discussion section: Contains a lot of background information but lacks a strong synthesis of what this new study adds to the understanding of PA pathogenesis. The discussion section should be broken up into small sections focused on a topic, and could also be revised to be about half of its current length.

Line 903-906: Again, you can't claim a 'significant' increase in ROS levels that you measured only in relative values.

Reviewer #2 (Comments for the Author):

The study conducted by Ren et al. shows the role of *lasI* and *rhlI* genes in conferring pathogenicity to *P. aeruginosa* by affecting several genes involved in quorum sensing pathway. Given the lack of sufficient literature on the pathways involved in quorum sensing system, this study provides novel insights to the subject matter. Overall, the manuscript is well-written and provides sufficient details in every section. Authors conducted several through experiments to characterize the study genes. My specific comments are provided below.

1. Some of the method section details are repeated in the results section. For instance L439-446 and L454-462. This text needs to be very brief to avoid repetition.

2. Authors are presenting some concluding statements in the results section. Please leave these out for the discussion. For instance L644-646 and L680-682.

3. The major concern is regarding the statistical analysis. ANOVA tables are not provided for any of the tests. Please provide these tables. These can also be added in supplementary info. Also, no information is provided on the replicates. Please add the number of replicates used for each test in the methods section, figure captions as well as table captions.

Ren et al. report on findings that the *lasI* and *rhlI* genes are involved in pathogenic behaviors of the widespread bacterial pathogen *Pseudomonas aeruginosa*. This research manuscript described data using an array of assays to investigate the behavior of mutants lacking one or both of these genes relative to that of the WT. The figures are well put together and the data support most of the author's conclusion. The primary thing lacking in this work is an organized narrative that synthesizes what is new about this work. For this reason, I recommend a concluding figure with a model of their system of interest that depicts how the author's data have expanded understanding of quorum-sensing in PA pathogenesis. The author major concern is that throughout the text (for example Line 84-90) the authors vastly overstate the importance of their findings. Their assays are in cell culture, and they do not directly test virulence or infection. The authors extrapolate from their in vitro studies that biofilm formation, and some other bacterial systems, play a "prominent role in attenuating virulence." Moreover, these in vitro assays were performed under normoxic conditions with no control for atmosphere, despite the hypoxic conditions in which PA infections naturally experiences in the lung and soft tissue infections. Thus, these claims (such as "targeting the quorum sensing system may be an ideal option for infection control and treatment") must be softened and brought into line with their evidence.

Other concerns are below.

The authors do not employ the use of complement strains, in which *lasI*, *rhlI* are added back. Thus, they cannot exclude the possibility that some of the phenotypes they observe are either from additional culturing and domestication of the mutant strains. This may be relevant for some of the milder phenotypes, for instance the bacteria adhesion assay timecourse, which shows little difference between the strains (5A).

Line 59. The authors should explain and put into context what quorum sensing is, and why it plays a role in pathogenesis.

Line 63. Similarly, the authors must explain why these mutants relate to quorum sensing.

Line 63. The relevance of THP-1 macrophages to *P. aeruginosa* pathogenesis is not explained.

Line 309-316. Measuring Congo Red dye is *not* a measure of c-di-GMP, but a measure of EPS.

Line 438-466 (and elsewhere): Keep methods in the methods section. Do not describe in detail bacterial mutant generation in the results. This also does not require a main text figure (1 &2), though that could be included in the supplemental.

Line 541-546: These microscopy analyses should be quantified, with multiple replicates, to justify these claims. You cannot draw this conclusion from single images alone.

Fig. 6B: You should denote the band of interest with a marking on the image.

Line 576. Incorrect. You did not visualize c-di-GMP levels by staining for EPS. You infer c-di-GMP levels from this assay. Please revise to be correct.

Line 582-594: These are rather archaic methods to analyze bacterial motility. I recommend using microscopy and live imaging of these behaviors, as is standard practice now in the study of bacterial motility. Further, the type of motility being observed is often incorrectly attributed through these types of plate/agar-based assays.

Line 616: What is a 'redox ability' ?

Line 631: Why did you select to work with macrophages? My understanding is that it is neutrophils that play a more dominant role in clearing PA infection (<https://www.ncbi.nlm.nih.gov/pmc/articles/PMC5371898/>). These cells have very different ROS-generation behavior and abilities (NETosis, for example).

Line 683 and Fig. 11: Without quantitative data on ROS concentrations you cannot say whether or not "PA01 induces a significant oxygen burst in macrophages...can alleviate this increased oxidative stress state." ROS have healthy normal levels that are part of normal eukaryotic cell processes. Most cells, bacterial and eukaryotic, are also exceptionally well-equipped to eliminate even high levels of ROS through catalase and peroxiredoxin enzymes. Hence, 'oxidative stress' needs to be proven by a method such as quantifying protein carbonylation or DNA damage. The authors have not done so, and should limit their claims to relative increases or decreases of fluorescence. Moreover, it isn't clear there is a large difference in the images presented in Fig. 11A. Can the authors confirm that the images shown were captured with identical background and thresholding levels?

Discussion section: Contains a lot of background information but lacks a strong synthesis of what this new study adds to the understanding of PA pathogenesis. The discussion section should be broken up into small sections focused on a topic, and could also be revised to be about half of its current length.

Line 903-906: Again, you can't claim a 'significant' increase in ROS levels that you measured only in relative values.

Journal: Microbiology Spectrum

Manuscript Number: Spectrum04146-23

Title: Mutation of *Pseudomonas aeruginosa lasI/rhlI* diminishes its cytotoxicity, inflammatory response and oxidative stress on THP-1 macrophages

Dear Dr. Dhammika Navarathna and Reviewers,

On behalf of myself and my co-authors, we would like to express our sincere thanks for the helpful feedback and suggestions.

Following your invaluable comments, we have diligently revised the manuscript and meticulously reorganized the language in a thorough and detailed manner. The specific corrections and revisions have been incorporated into our manuscript and **highlighted in red**. The attached document is the revised version that we would like to submit to the journal for your consideration.

In addition, we have prepared a comprehensive document entitled “Response to editor and reviewers”, which includes our detailed responses to the editor’s and reviewers’ comments, as well as our major revisions.

We sincerely hope that this revised manuscript will meet your standards and be approved. Thank you very much for your hard work! we are looking forward to hearing from you.

Best regards,

Yours sincerely,

Xinwei Liu

Corresponding author: Xinwei Liu (Email: 43154727@qq.com)

Responses to the comments and suggestions from the editor and reviewers

Dear Dr. Dhammika Navarathna and Reviewers,

On behalf of my co-authors, we deeply appreciate your positive and constructive comments and suggestions on our manuscript entitled “Mutation of *Pseudomonas aeruginosa* lasI/rhII diminishes its cytotoxicity, inflammatory response and oxidative stress on THP-1 macrophages” (Spectrum04146-23). We have carefully revised our manuscript according to the comments provided by the editor and reviewers, which we hope will meet with approval. Revised sections are **marked in red** in the revised manuscript. The major corrections and the responses to the editor’s and reviewers’ comments are as follows.

Responses to the reviewers’ comments:

To Reviewer #1:

Comment 1: Ren et al. report on findings that the lasI and rhII genes are involved in the pathogenic behaviors of the widespread bacterial pathogen *Pseudomonas aeruginosa*. This research manuscript described data using an array of assays to investigate the behavior of mutants lacking one or both of these genes relative to that of the WT. The figures are well put together and the data support most of the author’s conclusion. The primary thing lacking in this work is an organized narrative that synthesizes what is new about this work. For this reason, I recommend a concluding figure with a model of their system of interest that depicts how the author’s data have expanded their understanding of quorum-sensing in PA pathogenesis. The author major concern is that throughout the text (for example Line 84-90) the authors vastly overstate the importance of their findings. Their assays are in cell culture, and they do not directly test virulence or infection. The authors extrapolate from their in vitro studies that biofilm formation, and some other bacterial systems, play a "prominent role in attenuating virulence." Moreover, these in vitro assays were performed under

normoxic conditions with no control for atmosphere, despite the hypoxic conditions in PA infections naturally experiences in the lung and soft tissue infections. Thus, these claims (such as "targeting the quorum sensing system may be an ideal option for infection control and treatment") must be softened and brought into line with their evidence.

Response: Thanks for your valuable suggestions, which greatly enhanced the novelty and readability of our research. Following your advice, we reorganized the narrative in the manuscript and a systematic conclusion graph was drew to express our work more clearly. We have included these modifications in our manuscript, and the model figure are presented in **FIG. 11**.

In addition, thanks for your helpful reminder regarding the "Importance" section. Considering the limitations of our study focusing only on macrophages and the conditions under which we performed the experiments *in vitro*, we amend our claims in Line 88-90 "This finding suggests that effective intervention targeting the quorum sensing system may be an ideal option for infection control and treatment" to "**These findings suggest that the effective intervention targeting the quorum sensing system *lasI/rhlI* is expected to reduce the pathogenicity of *Pseudomonas aeruginosa* to THP-1 macrophages**" in **Lines 95-97 on Page 5** to make the importance consistent with our findings.

Other comments:

Comment 2: The authors do not employ the use of complement strains, in which *lasI*, *rhlI* are added back. Thus, they cannot exclude the possibility that some of the phenotypes they observe are either from additional culturing and domestication of the mutant strains. This may be relevant for some of the milder phenotypes, for instance, the bacteria adhesion assay time course, which shows little difference between the strains (5A).

Response: We sincerely thank you for your valuable suggestions, which helped improve the quality of this paper. The construction of complement strains is already in

our plan. However, due to the time limit, we sincerely apologize for not obtaining the complement strains on time, which is a drawback of this study. Nevertheless, we strive to enhance other aspects of our work to meet the requirements for article quality. In future studies, the complement strains will be generated, and animal infection models will be constructed to investigate the effect of *lasI/rhlI* deletion in the QS system on the pathogenicity of *P. aeruginosa* from the perspective of infection *in vivo*.

Comment 3: Line 59. The authors should explain and put into context what quorum sensing is, and why it plays a role in pathogenesis.

Response: Thank you very much for your helpful suggestions. We have added an explanation in the abstract about what quorum sensing is and why it plays a role in pathogenesis. Our revisions can be found in **Lines 59-65** on **Page 4** of the “**Abstract**”. The specifics of our additions are highlighted in red below: “**Quorum sensing is an inter-bacterial communication mechanism that coordinates bacterial activities in response to changes in community density. Quorum sensing systems regulate a diverse array of virulence involved in various mechanisms, including robbing host nutrients, providing scaffolds for biofilm formation, aiding motility, producing virulence factors, and providing protection against host immune attack, which plays a crucial role in colonization and infection for *Pseudomonas aeruginosa*”.**

Comment 4: Line 63. Similarly, the authors must explain why these mutants relate to quorum sensing.

Response: Thank you very much for your kind reminder. We have listed the reason why these mutants relate to quorum sensing in **Lines 65-66** on **Page 4** of the “**Abstract**” section. Specifically: “**The LasI/R and RhI/R sub-systems dominate in the quorum sensing system of *Pseudomonas aeruginosa*”.**

Comment 5: Line 309-316. Measuring Congo Red dye is not a measure of c-di-GMP, but a measure of EPS.

Line 576. Incorrect. You did not visualize c-di-GMP levels by staining for EPS. You infer c-di-GMP levels from this assay. Please revise to be correct.

Response: Thank you for your rigorous and professional review, which is very valuable to us. We apologize for the mistake. We have updated the method to this section of “**Exopolysaccharide assay**” in the “**MATERIALS AND METHODS**”, and the corresponding results and figure numbers have also been corrected.

Comment 6: Line 438-466 (and elsewhere): Keep methods in the methods section. Do not describe in detail bacterial mutant generation in the results. This also does not require a main text figure (1 &2), though that could be included in the supplemental.

Response: Thank you for your helpful suggestions. We have removed Figures 1 & 2 and their corresponding resulting descriptions from the main text and included them in the “**Supplementary Material**” as **FIG S1** and **FIG S2**.

Comment 7: Line 541-546: These microscopy analyses should be quantified, with multiple replicates, to justify these claims. You cannot draw this conclusion from single images alone.

Response: Thank you very much for your valuable advice. It has helped us a lot to verify our claims. Following the suggestion, we experimented with triplicate, and the mean fluorescence intensity was quantified using ImageJ software. The corresponding **statistical analysis result** is added in **FIG 3E**.

Comment 8: Fig. 6B: You should denote the band of interest with a marking on the image.

Response: Thank you very much for your kind reminder. We have marked a protein band at 33 kDa in updated **Fig. 4B**.

Comment 9: Line 582-594: These are rather archaic methods to analyze bacterial motility. I recommend using microscopy and live imaging of these behaviors, as is

standard practice now in the study of bacterial motility. Further, the type of motility being observed is often incorrectly attributed through these types of plate/agar-based assays.

Response: Thank you very much for your professional suggestions, which help enhance the novelty and scientific merit of the manuscript. We have included microscope real-time motion imaging in the method of “**Bacterial motility**”, on **page 13**. Corresponding analysis results are added to the results of “**Mutation of *lasI/rhlI* attenuates the bacterial motility of *P. aeruginosa***” on **page 21** and presented in **FIG 5G-I**, the corresponding videos are included in **Video S1-S3** of “**Supplemental material**”.

In addition, taking into account the differences in bacterial motility in AGAR and liquid, we retained the results of the AGAR plate movement experiment, to investigate bacterial motility more comprehensively.

Comment 10: Line 616: What is a 'redox ability'?

Response: Thank you very much for your helpful question. Our description here may be vague, pyocyanin is a green pigment produced by *P. aeruginosa*, it can undergo reversible REDOX action, oxidation type is blue (alkaline) and red (acidic), $pK=5.0$ at the color change point; The reductants are colorless but appear in an intermediate stage of REDOX under acidic conditions (green). REDOX potential $E_0' = -0.034V(pH=7)$. Thus, we have revised “Pyocyanin is a phenazine compound with strong redox ability secreted by *P. aeruginosa*” to “**Pyocyanin is a phenazine compound with reversible REDOX activity secreted by *P. aeruginosa***” and was shown in **Lines 572-573** on **Page 22** of “**Results**” section. Furthermore, a more detailed explanation is included in the “**Discussion**” section, was shown in **Lines 722-725** on **Page 27** as: “**Pyocyanin is a REDOX-active factor and is crucial in the virulence and pathogenicity of *P. aeruginosa* infection by inhibiting cellular respiration, disrupting intracellular REDOX balance, inducing host damage, and contributing to immune evasion**”.

The references are listed below:

- Hall S, McDermott C, Anoopkumar-Dukie S, McFarland AJ, Forbes A, Perkins AV, Davey AK, Chess-Williams R, Kiefel MJ, Arora D, Grant GD. Cellular Effects of Pyocyanin, a Secreted Virulence Factor of *Pseudomonas aeruginosa*. *Toxins* (Basel). 2016 Aug 9;8(8):236. doi: 10.3390/toxins8080236.
- Ho Sui SJ, Lo R, Fernandes AR, Caulfield MD, Lerman JA, Xie L, Bourne PE, Baillie DL, Brinkman FS. 2012. Raloxifene attenuates *Pseudomonas aeruginosa* pyocyanin production and virulence. *Int J Antimicrob Agents* 40:246-51. doi:10.1016/j.ijantimicag.2012.05.009.
- Bastos RW, Akiyama D, Dos Reis TF, Colabardini AC, Luperini RS, de Castro PA, Baldini RL, Fill T, Goldman GH. 2022. Secondary Metabolites Produced during *Aspergillus fumigatus* and *Pseudomonas aeruginosa* Biofilm Formation. *mBio* 13: e0185022. doi:10.1128/mbio.01850-22.

Comment 11: Why did you select to work with macrophages? My understanding is that it is neutrophils that play a more dominant role in clearing PA infection (<https://www.ncbi.nlm.nih.gov/pmc/articles/PMC5371898/>). These cells have very different ROS-generation behavior and abilities (NETosis, for example).

Response: Many thanks for your constructive questions. In our consideration, “Macrophages play a crucial role in the innate immune response, actively phagocytosing and eliminating invading pathogens during the early infection stages when the bacterial load is insufficient to recruit neutrophils”. Therefore, we propose that macrophages play a pivotal role in the early stage of immune phagocytosis during *Pseudomonas aeruginosa* invasion, which is crucial for effectively clearing acute *Pseudomonas aeruginosa* infection and potentially preventing the development of chronic infection. We have included this description in the “**INTRODUCTION**” section in lines 131-133 on Page 7, and hope that our revisions are appropriate and will be recognized.

The reference is listed below:

Neupane AS, Willson M, Chojnacki AK, Vargas ESCF, Morehouse C, Carestia A, Keller AE, Peiseler M, DiGiandomenico A, Kelly MM, Amrein M, Jenne C, Thanabalasuriar A, Kubes P. 2020. Patrolling Alveolar Macrophages Conceal Bacteria from the Immune System to Maintain Homeostasis. *Cell* 183:110-125.e11. doi: 10.1016/j.cell.2020.08.020.

Comment 12: Line 683 and Fig. 11: Without quantitative data on ROS concentrations you cannot say whether or not "PA01 induces a significant oxygen burst in macrophages...can alleviate this increased oxidative stress state." ROS have healthy normal levels that are part of normal eukaryotic cell processes. Most cells, bacterial and eukaryotic, are also exceptionally well-equipped to eliminate even high levels of ROS through catalase and peroxiredoxin enzymes. Hence, 'oxidative stress' needs to be proven by a method such as quantifying protein carbonylation or DNA damage. The authors have not done so, and should limit their claims to relative increases or decreases of fluorescence. Moreover, it isn't clear there is a large difference in the images presented in Fig. 11A. Can the authors confirm that the images shown were captured with identical background and thresholding levels?

Response: Thank you for your valuable suggestions, which are of great significance in enhancing the scientific merit of this study. Following the recommendations, we have supplemented the experiments as follows.

- 1) To further validate the oxidative stress, we quantified the level of protein carbonylation to assess protein oxidative damage.
- 2) Additionally, Western blot and RT-qPCR were employed to detect the protein and mRNA expression levels of antioxidant enzymes SOD2 and GPX4.
- 3) For inverted fluorescence microscopy of intracellular ROS levels, the assay was re-run to obtain a clearer image.
- 4) Furthermore, flow cytometry detection of ROS levels was also performed using

Flowjo_v10.9.0 software to redraw the overlay plots, the mean fluorescence intensity values of ROS were subjected to statistical analysis.

All the above results have been shown in “**Mutation of lasI/rhlI alleviates the oxidative stress of THP-1 macrophages induced by *P. aeruginosa***” of “**Results**” in **Lines 606-623** on **Pages 23-24**, and the figures were consolidated into **FIG. 8**.

- 5) In addition, for DNA damage, we employed the TUNEL kit to detect DNA breaks.
- 6) Based on the observed phenomenon of DNA damage and its promotion in apoptosis, we subsequently conducted an Annexin V-FITC/PI apoptosis assay using flow cytometry.
- 7) We also assessed the relative protein expression levels of Bax and Bcl-2 proteins in the apoptosis/anti-apoptosis system using Western blot.

This section has been added in the manuscript as “**Mutation of lasI/rhlI reduces DNA damage and apoptosis of THP-1 macrophages induced by *P. aeruginosa***” of “**Results**” section in **Lines 644-663** on **Page 25**, and the figures were consolidated into **FIG. 10**.

Comment 13: Discussion section: Contains a lot of background information but lacks a strong synthesis of what this new study adds to the understanding of PA pathogenesis. The discussion section should be broken up into small sections focused on a topic, and could also be revised to be about half of its current length.

Response: Thank you for your meticulous comments. To highlight our research objectives and clarify the significance of this study in the pathogenesis of *Pseudomonas aeruginosa*, we have resorted and analyzed the discussion section comprehensively to make the discussion more compact and profound. We hope that our revisions are appropriate and will be recognized.

Comment 14: Line 903-906: Again, you can't claim a 'significant' increase in ROS

levels that you measured only in relative values.

Response: Thank you for your valuable comments. To substantiate our claim, we have incorporated the research findings on protein carbonylation and DNA damage, in addition to solely detecting intracellular ROS levels as described in the original manuscript. The mean fluorescence intensity values of ROS were obtained from flow cytometry. Furthermore, we have assessed the expression of antioxidant enzymes SOD2 and GPX4 at both protein and transcriptional levels.

To Reviewer #2:

The study conducted by Ren et al. shows the role of *lasI* and *rhlI* genes in conferring pathogenicity to *P. aeruginosa* by affecting several genes involved in the quorum sensing pathway. Given the lack of sufficient literature on the pathways involved in quorum sensing systems, this study provides novel insights into the subject matter. Overall, the manuscript is well-written and provides sufficient details in every section. The authors conducted several thorough experiments to characterize the study genes. My specific comments are provided below.

Comment 1: Some of the method section details are repeated in the results section. For instance L439-446 and L454-462. This text needs to be very brief to avoid repetition.

Response: We appreciate your valuable feedback. To enhance the professionalism of the narration in our manuscript, we have excluded the section about the construction of *ΔlasI* and *ΔlasIIΔrhlI* mutant strains from the “Results” section and included it as **FIG S1** and **FIG S2** in the “**Supplementary material**” file. Additionally, we have meticulously refined other parts of the results section to avoid repetition.

Comment 2: Authors are presenting some concluding statements in the results section. Please leave these out for the discussion. For instance L644-646 and L680-

682.

Response: Thank you for your helpful advice. We have thoroughly examined the full text, excluding the concluding statements in the results section, and subsequently arranged them within the discussion section. The discussion has been rearranged to ensure a coherent narrative that adheres to academic standards and aligns with the requirements of the Microbiology Spectrum journal.

Comment 3: The major concern is regarding the statistical analysis. ANOVA tables are not provided for any of the tests. Please provide these tables. These can also be added in supplementary info. Also, no information is provided on the replicates. Please add the number of replicates used for each test in the methods section, figure captions as well as table captions.

Response: Thank you for your valuable suggestion. We have included tables of partial ANOVA statistics in the “**Supplementary Material**” as **Table S5-S8**. Furthermore, we have provided detailed information regarding the number of replicates utilized for each test in the methods section, figure headings, and table headings.

We have made every effort to improve the manuscript and implemented major revisions. These changes do not affect the content or framework of the paper. We sincerely appreciate the valuable suggestions from both the editor and the reviewers and hope that the revised manuscript will meet the publication standards of the journal and be accepted for publication.

Once again, thank you very much for your insightful comments and suggestions. We are looking forward to hearing from you.

With best wishes,

Yours sincerely,

Yanying Ren, Xiaojuan You, Rui Zhu, Dengzhou Li, Chunxia Wang, Zhiqiang He,
Yue Hu, Yifan Li, Xinwei Liu, Yongwei Li

Corresponding author: Xinwei Liu (Email: 43154727@qq.com)

Re: Spectrum04146-23R1 (Mutation of *Pseudomonas aeruginosa* lasI/rhlI diminishes its cytotoxicity, inflammatory response and oxidative stress on THP-1 macrophages)

Dear Dr. Xinwei Liu:

Thank you for the privilege of reviewing your work. Below you will find my comments, instructions from the Spectrum editorial office, and the reviewer comments.

Gene complementation is essential to ensure the quality of this work and at present day science, I will give another chance for the authors to complement the gene and report results.

Revision Guidelines

Sincerely,
Dhammika Navarathna
Editor
Microbiology Spectrum

Reviewer #1 (Comments for the Author):

I have no further comments on this work.

No further comments.

Journal: Microbiology Spectrum

Manuscript Number: Spectrum04146-23

Title: Mutation of *Pseudomonas aeruginosa lasI/rhlI* diminishes its cytotoxicity, inflammation and oxidative stress on THP-1 macrophages

Dear Dhammika Navarathna Editor,

On behalf of myself and my co-authors, we would like to express our sincere thanks for your valuable feedback on enhancing this manuscript's quality and scientific nature.

After receiving the suggestion on gene complementation of mutant strains, we have intensified our work process to meet the deadline. We completed the complementary work of the two mutant strains and supplemented the experimental results of complemented strains in all experiments. The corresponding experimental results have been updated and included in the revised manuscript, supplementary materials, and figures.

We have made every effort to improve the manuscript, and these changes do not affect the overall framework of the paper. We sincerely hope this revised manuscript will meet your standards and be approved. Thank you very much for your hard work. We are looking forward to hearing from you.

Best regards,

Yours sincerely,

Xinwei Liu (Email: 43154727@qq.com)

Re: Spectrum04146-23R2 (Mutation of *Pseudomonas aeruginosa* lasI/rhlI diminishes its cytotoxicity, inflammatory response and oxidative stress on THP-1 macrophages)

Dear Dr. Xinwei Liu:

Thank you for the privilege of reviewing your work. Below you will find my comments, instructions from the Spectrum editorial office, and the reviewer comments.

Please respond the concerns # 2 reviewer raised in the previous version.

Revision Guidelines

Sincerely,
Dhammika Navarathna
Editor
Microbiology Spectrum

Reviewer #1 (Comments for the Author):

The authors have worked to improve the manuscript presentation, writing, and experiments. I have no further comments on this manuscript.

Reviewer #4 (Comments for the Author):

This is the second revision to the original manuscript. Authors addressed the issue of showing the results of complemented strains. However, there are several points to revise (see below).

- I still consider that the Results shown in Figs 1 to 6 lack novelty, and they were already reported in the literature by other researchers. Authors should shorten this part (may be by sending most to the Figs to supplemental material).
- Revise English in the entire manuscript
- Abstract is too long; it should be shortened.
- All along the manuscript, except the first time that the bacterium is mentioned, *P. aeruginosa* should be written instead of *Pseudomonas aeruginosa*.

The authors have worked to improve the manuscript presentation, writing, and experiments. I have not further comments on this manuscript.

Journal: Microbiology Spectrum

Manuscript Number: Spectrum04146-23

Title: Mutation of *Pseudomonas aeruginosa lasI/rhlI* diminishes its cytotoxicity, inflammation and oxidative stress on THP-1 macrophages

Dear Dr. Dhammika Navarathna and Reviewer #2,

On behalf of myself and my co-authors, we would like to express our heartfelt gratitude for your valuable feedback and suggestions on improving the quality and novelty of this manuscript.

Following your invaluable comments, we have thoroughly revised the manuscript and meticulously reorganized the figures and language in detail. The specific corrections and revisions have been updated and included in the newly uploaded manuscript, supplementary material, and figures. The major revisions and responses to the comments of Reviewer #2 are represented below.

We have made every effort to improve the manuscript and these changes do not alter the overall framework of the paper. We sincerely hope that this revised version will meet your standards and be approved. Thank you very much for your hard work, we are looking forward to hearing from you.

Best regards,

Yours sincerely,

Xinwei Liu (Email: 43154727@qq.com)

Responses to Reviewer #2's comments:

Comment 1: I still consider that the Results shown in Figs 1 to 6 lack novelty, and they were already reported in the literature by other researchers. Authors should shorten this part (maybe by sending most to the Figs to supplemental material).

Response: Thank you very much for your valuable comment, which helps enhance the novelty of the manuscript.

Following your suggestions, we have transferred most of the original Fig. 1-6 to the supplementary material. Specifically, we have moved the previous Fig. 1-2 to Fig. S3-S4. The previous Fig. 3-6 depict various aspects of bacterial adhesion, biofilm formation, EPS production, motility and virulence factors. To align with the model figure represented in the current Fig. 7, we have retained partial key results in each major category of experiments and reorganized them into the current Fig. 1-2, while the remaining results are now included in the supplementary material as Fig. S5-6. The corresponding descriptions have also been updated in the manuscript accordingly. We sincerely hope that our presentation and layout are appropriate and appealing.

Comment 2: Revise English in the entire manuscript.

Response: We are grateful for your valuable suggestion, which helps improve the readability of this paper. We have revised the English throughout the entire manuscript meticulously and carefully, and we hope that our revisions will be approved.

Comment 3: Abstract is too long; it should be shortened.

Response: We sincerely appreciate your valuable suggestion. We have shortened and refined the abstract from 311 words to 243 words to meet publication standards.

Comment 4: All along the manuscript, except the first time that the bacterium is mentioned, *P. aeruginosa* should be written instead of *Pseudomonas aeruginosa*.

Response: Thank you very much for your kind reminder. We have revised the term *Pseudomonas aeruginosa* to *P. aeruginosa*, except for its first mention in the

manuscript.

Re: Spectrum04146-23R3 (Mutation of *Pseudomonas aeruginosa* lasI/rhlI diminishes its cytotoxicity, oxidative stress, inflammation and apoptosis on THP-1 macrophages)

Dear Dr. Xinwei Liu:

Your manuscript has been accepted, and I am forwarding it to the ASM production staff for publication. Your paper will first be checked to make sure all elements meet the technical requirements. ASM staff will contact you if anything needs to be revised before copyediting and production can begin. Otherwise, you will be notified when your proofs are ready to be viewed.

Sincerely,
Dhammika Navarathna
Editor
Microbiology Spectrum

Reviewer #4 (Comments for the Author):

This revised version addressed the concerns I raised previously.